# Heterogeneous changes of soil microclimate in high mountains and glacier forelands

Silvio Marta ®[1,2] ✉, Anaïs Zimmer ®[3], Marco Caccianiga ®[4], Mauro Gobbi ®[5], Roberto Ambrosini[1], Roberto Sergio Azzoni[1,6], Fabrizio Gili ®[1,7], Francesca Pittino[8], Wilfried Thuiller[9], Antonello Provenzale[2] & Gentile Francesco Ficetola ®[1,9]

Landscapes nearby glaciers are disproportionally affected by climate change, but we lack detailed information on microclimate variations that can modulate the impacts of global warming on proglacial ecosystems and their biodiversity. Here, we use near-subsurface soil temperatures in 175 stations from polar, equatorial and alpine glacier forelands to generate high-resolution temperature reconstructions, assess spatial variability in microclimate change from 2001 to 2020, and estimate whether microclimate heterogeneity might buffer the severity of warming trends. Temporal changes in microclimate are tightly linked to broad-scale conditions, but the rate of local warming shows great spatial heterogeneity, with faster warming nearby glaciers and during the warm season, and an extension of the snow-free season. Still, most of the fine-scale spatial variability of microclimate is one-to-ten times larger than the temporal change experienced during the past 20 years, indicating the potential for microclimate to buffer climate change, possibly allowing organisms to withstand, at least temporarily, the effects of warming.

Mountain ecosystems provide multiple goods and services to humankind and act as fundamental regulators of regional climate and hydrology[1–3]. The topographic and climatic heterogeneity of mountain areas, as well as their geological history, deeply influence several biological processes (i.e. adaptation, speciation, dispersal, persistence, and extinction[4]); as a result, mountain ecosystems are biodiversity hotspots with unique levels of endemism, adaptations and lifeforms[5]. For instance, despite mountain regions covering only one-fourth of the Earth continental surface (excluding Antarctica), they host > 80% of the world's species of terrestrial vertebrates, many of which are entirely restricted to mountains[5–7]. However, ongoing climatic changes are causing unprecedented modifications of mountain systems[2,3]. At the highest elevations, glaciers are losing mass, and the pace of glacier

retreat has been globally accelerating during the past decades[8]. This dramatic glacier shrinkage has multiple impacts on all biotic and abiotic components of ecosystems[3,9–12]. Glacier forelands are new landscapes emerging after the retreat of glaciers[3] that undergo rapid geomorphological transformations, with loose sediments from the early-successional stages rapidly developing into structured soils[9,13,14]. In turn, soil development facilitates the colonization of recently deglaciated terrains by multiple lifeforms[12,15]. However, climatic variations affect the rates of change of these ecosystems. For instance, temperature influences the rate of rock weathering[16,17] and warmer areas experience faster soil development[14]. High temperatures can also affect temporal dynamics of communities, by increasing colonization success by termophilic species and favouring evolution towards more

---

[1]Department of Environmental Science and Policy, University of Milan, Via G. Celoria 10, 20133 Milan, Italy. [2]Institute of Geosciences and Earth Resources, IGG-CNR, Italian National Research Council, 56124 Pisa, Italy. [3]Department of Geography and the Environment, University of Texas at Austin, 78712 Austin, TX, USA. [4]Department of Biosciences, University of Milan, via G. Celoria 26, 20133 Milan, Italy. [5]Research & Museum Collections Office, Climate and Ecology Unit, MUSE-Science Museum, Corso del Lavoro e della Scienza 3, 38122 Trento, Italy. [6]Department of Earth Sciences "Ardito Desio", University of Milan, Via L. Mangiagalli 34, 20133 Milan, Italy. [7]Department of Life Sciences and Systems Biology, University of Turin, Via Accademia Albertina 13, 10123 Turin, Italy. [8]Department of Earth and Environmental Sciences (DISAT) - University of Milan-Bicocca, Milan, Italy. [9]Univ. Grenoble Alpes, Univ. Savoie Mont Blanc, CNRS, LECA, F38000 Grenoble, France. ✉e-mail: silvio.marta@hotmail.it

complex community structures[18–20], and influence carbon fluxes between soil, vegetation and the atmosphere[21,22].

In areas with complex terrain, regional climate (i.e. macroclimate) interacts with topography, potentially resulting in local temperatures partially decoupled from the regional average[23]. Microclimate can be defined as the fine-scale spatial and temporal offsets of the local climate from the macroclimate[24]. The decoupling between micro- and macroclimate is particularly pronounced near and, to a lesser extent, below soil surface[25,26], where microclimate best represents the set of climatic conditions actually experienced by organisms. In mountain areas, topographic elements (i.e. elevation, mainly via lapse-rates; aspect; slope and topographic shading) locally regulate temperature, incoming solar radiation, evapotranspiration, wind speed, cold air drainage, and snow accumulation and melt at fine spatial scales, generating complex patterns of local climatic conditions[23,27,28]. Along glacier forelands, additional factors influence local climate, such as vegetation cover and height, distance from the ice mass, and soil texture, creating heterogeneous microhabitats inhabited by different biotic assemblages[29]. Snow cover is a further key driver of the functioning of mountain ecosystems[30], affecting biogeochemical and hydrological processes, and controlling the life cycle of many organisms by determining the duration of their growing / activity season[30–32], with potential impacts on ecosystem productivity[31].

At fine spatial scales, spatial variability of local temperature and snow can be strong, creating a mosaic of nearby micro-habitats that host different communities[31,33]. Microclimate differences between nearby areas might at least temporarily buffer the severity of warming impacts on populations. Microclimatic buffering is the dampening of macro-climatic fluctuations due to local conditions (e.g. topography and vegetation cover), such that larger-scale fluctuations still exist at the microclimatic scale, but have lower intensity and a reduced impact on organisms[34,35]. Clearly, microclimate heterogeneity can limit the impacts of macroclimate change only if organisms are able to move between neighbouring sites having different microclimates[35–38]. Detailed information on microclimate and snow cover is thus pivotal to understand the impacts of climate change on organisms living in mountain ecosystems and the potential buffering effect favoured by microclimatic heterogeneity. Still, this requires global scale, high-resolution analyses that were so far lacking.

Here, we used a unique dataset of near-subsurface soil temperatures collected in 175 stations from polar, equatorial and alpine glacier forelands to produce a high-resolution, global reconstruction of monthly average soil temperatures during the snow-free season (i.e., when snow cover is strongly reduced or absent) in high mountains and proglacial environments (Fig. 1). To combine the accuracy of empirically-calibrated relationships with the transferability of process-based models, we implemented a correlative hybrid approach based on the mechanistic understanding of the main drivers of microclimate[39,40]. Terms were introduced in the modelling framework to account for both the horizontal (elevation, topography, topographic shading, permafrost occurrence, katabatic winds, monthly frequency of snow-free days) and vertical (depth and tree cover) processes driving microclimate variability[41]. To account for inter-annual variability and temporal change, and produce dynamic estimates of soil microclimate, our model was combined with time-series for macroclimate, frequency of snow-free days, shortwave radiation and glacier forefront position. Empirically-estimated coefficients were validated with an external dataset[26], and then used to assess temporal variation of microclimatic conditions between 2001–2005 and 2016–2020. The comparison between these two periods allowed measuring long-term annual and seasonal microclimate variations, provided estimates of the global-scale buffering effects of microclimate, and revealed that recently deglaciated habitats (those closer to the glacier forefront) are experiencing a much faster microclimatic change compared to other high-elevation environments.

## Results

### Temperature modelling

The 175 microclimatic stations provided 706,810 temperature records in the period 2011–2021 (see Methods). Soil temperature was positively related to macroclimatic temperature (downscaled using elevational lapse rate, see Methods), downward shortwave solar radiation, frequency of snow-free days and distance from the glacier forefront (indicating an effect of katabatic winds), while negative relationships were found with depth of burial and tree cover. Statistically significant interactions showed that the increase of soil temperature with macroclimatic temperature and solar radiation was faster as the frequency of snow-free days increased (e.g. moving from spring to summer; Fig. 2a, b). The decrease of soil temperature with increasing depth depended on the frequency of snow-free days, deep soil being relatively warmer at the beginning and the end of the season (low frequency of snow-free days - sfd) and colder during summer, and shallower soil responding faster to air temperatures (Fig. 2c). We detected no significant effect of permafrost occurrence (Fig. 2d), of the interaction between tree cover and solar radiation, and of the interaction between the monthly frequency of snow-free days and the distance from the glacier (Supplementary Table 1). The model provided a very good fit to the training dataset and explained a very high portion of microclimatic variations ($R^2_m = 0.71$, $R^2_c = 0.85$).

Downscaled macroclimate, solar radiation and frequency of snow-free days were the strongest drivers of soil temperature, considering either variable importance scores (i.e. joint contribution of both additive and interactive terms; Supplementary Fig. 1a), or semi-partial $R^2$ (Supplementary Fig. 1b). Among the remaining predictors, depth of burying explained a substantial portion of the total variance in soil temperature, especially when interacting with the frequency of snow-free days, while the contribution of tree cover, distance from the glacier forefront and permafrost occurrence was small (Supplementary Fig. 1).

Soil temperature values, predicted by averaging model coefficients across the 26 leave-one-out models (Supplementary Table 2), were in good agreement with the observed ones, and returned excellent predictions of temperature in the glacier forelands used for internal validation (Fig. 2e; $wR^2 = 0.85$; $wMAE = 1.47\,°C$; $wRMSE = 1.84\,°C$). The high transferability of the model, and its good predictive power, were confirmed by the validation on the independent dataset (Fig. 2f; $wR^2 = 0.91$; $wMAE = 1.95\,°C$; $wRMSE = 2.73\,°C$).

To understand how adding variables besides macroclimate improves the prediction of soil temperature, for both the training and independent datasets we compared observed temperatures with i) those predicted by the complete model, and the time-series of two widely used climate products: ii) TerraClimate[42] and iii) CHELSA[43] (Supplementary Fig. 2). Our model limited the underestimation of soil temperature that occurs with macroclimatic products during cold periods (Supplementary Fig. 2d vs 2e-f) and outperformed both of traditional climate products in predicting soil temperature, in terms of variance explained ($wR^2 = 0.86$ and 0.91 vs 0.60 to 0.91), mean absolute error ($wMAE = 1.45$ and $1.95\,°C$ vs 2.37 to $3.19\,°C$) and root mean square error ($wRMSE = 1.85$ and $2.73\,°C$ vs 3.00 to $4.49\,°C$).

### Global projections

Building upon the high transferability of our model, we upscaled it at the global scale for the periods 2001–2005 and 2016–2020. During 2001–2020, we detected substantial temperature increases in North America, the Andes and the higher latitudes of the Eastern Palaearctic, as well as in the European Alps and some areas of the Himalayas (Fig. 3a; Supplementary Fig. 3). When looking at different latitudinal bands (Fig. 3b), the pattern of temperature change showed differences between the Inter-tropical zone, the Northern and the Southern hemispheres. Temperature increase was particularly marked in the Inter-tropical zone and the Southern hemisphere (mean ± sd:

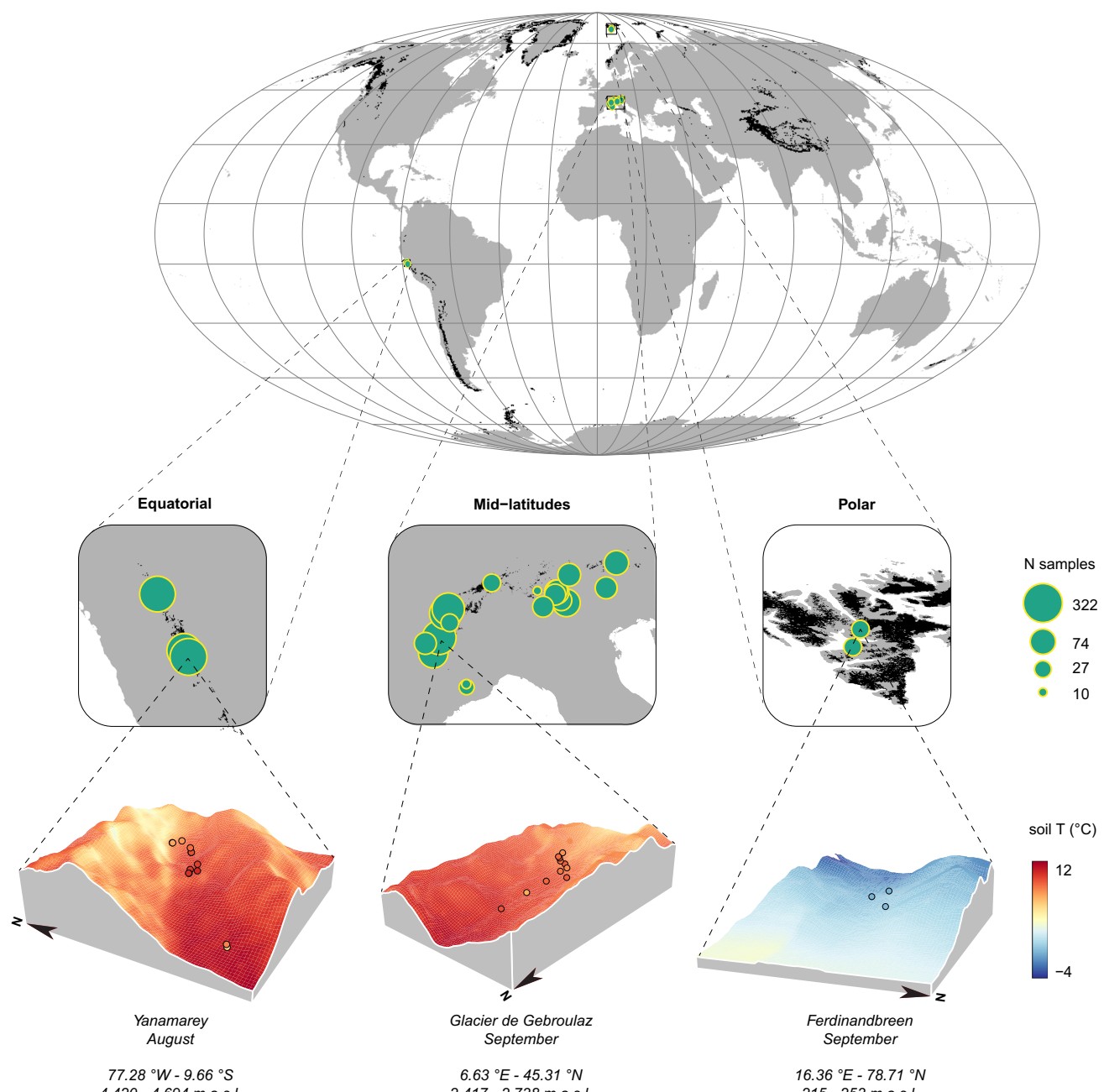

**Fig. 1 | Study area, distribution of the recording sites and examples of model output.** Black areas in the map and insets represent glaciers (source: GLIMS database[73]); green dot size is proportional to the per-glacier number of sampled months (loggers × months). For each example glacier, we report the coordinates of the centroid of the recording sites (EPSG:4326), the altitudinal range of the mapped area, and the observed (dots: August 2020 - Yanamarey; September 2019 - Glacier de Gebroulaz; September 2018 - Ferdinandbreen) and estimated (averaged over the period 2015–2019) soil temperature in a specific month. Map projection: Mollweide (ESRI:54009); grid: 20*20 degrees. Source data are provided as a Source Data file.

0.75 ± 0.53 and 1.02 ± 0.88 °C, respectively), with a generally higher increase nearby glaciers. For instance, in the Inter-tropical zone the mean increase was 1.13 ± 0.70 °C within 100 m from the glacier outline, while it was 0.57 ± 0.31 °C at 3 km from glaciers. The change was smaller in the Northern hemisphere (0.43 ± 0.61 °C), still temperature increase remained higher nearby glaciers compared to areas located 3 km away from the glacier (0.63 ± 0.84 °C vs. 0.34 ± 0.44 °C; Fig. 3b). These temperature changes were not identical to macroclimatic observations of climate change. The difference between microclimate and macroclimate was distinctly larger and more positive nearby glaciers and in tropical regions, highlighting the particularly fast warming of such areas (Supplementary Fig. 4).

The analysis of seasonal trends provided comparable results, in terms of spatial patterns of temperature changes (Supplementary Fig. 5). In the mountain ranges of the Northern hemisphere, the strongest temperature increase occurred during September-November, with a particularly intense increase at the higher latitudes (Supplementary Fig. 5b–d). In the Southern hemisphere, temperature changes were especially strong from December to February (Supplementary Fig. 5). Conversely, in the Inter-tropical zone temperature change was homogeneously distributed throughout the year, owing to the reduced effect of seasonality. A decrease of changes with increasing distance from the glacier was evident during the period September–February (Supplementary Fig. 6a, d). Seasonal trends

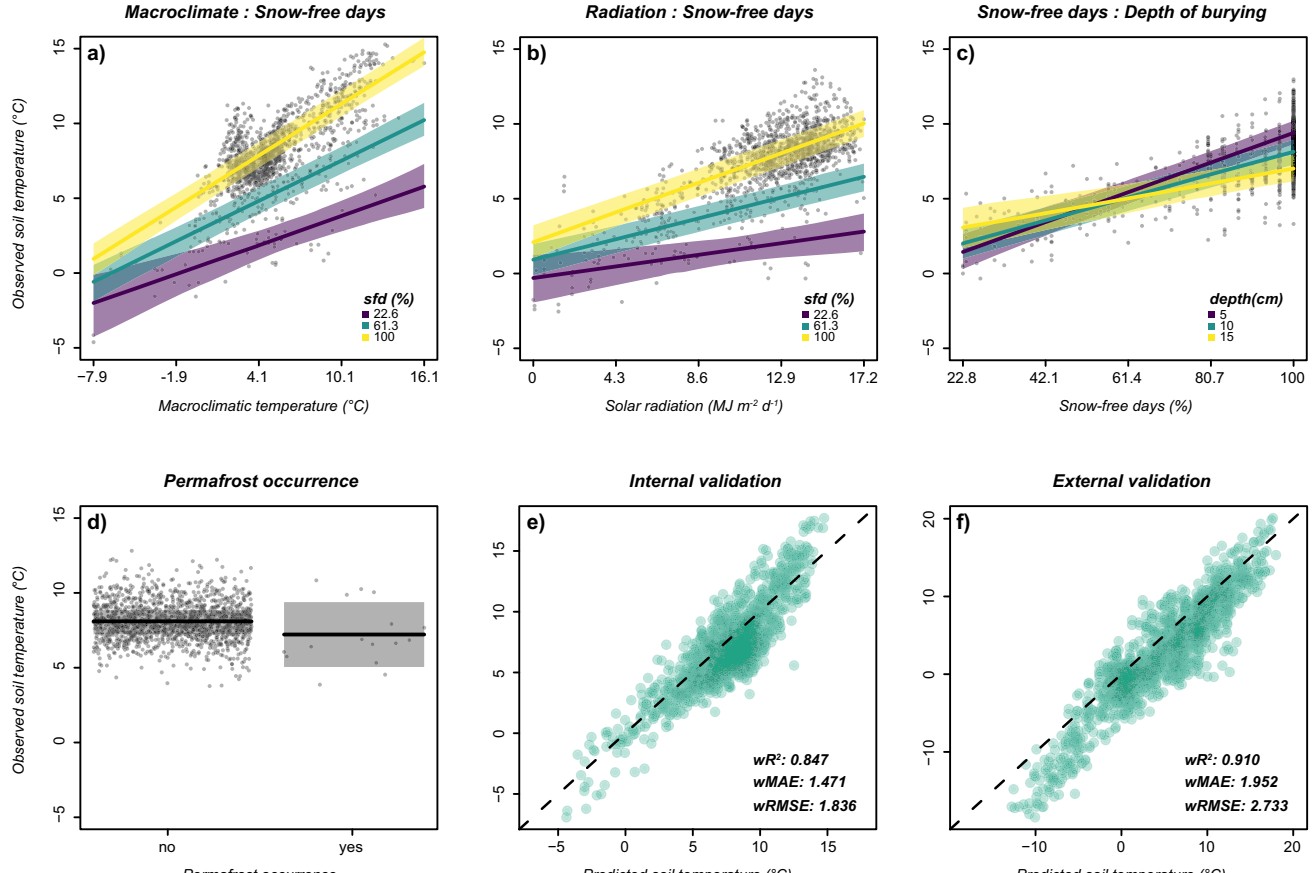

**Fig. 2 | Relationships between environmental predictors and soil temperature.**
Effect of (**a**) macroclimate, and (**b**) cumulative daily shortwave solar radiation at different monthly frequencies of snow-free days (sfd). Effects of (**c**) monthly frequency of snow-free days at different depths of burying (depth) and (**d**) permafrost occurrence. **e**, **f** Comparison between recorded soil temperatures and those predicted using mean coefficients from the leave-one-out analysis. **e** Internal validation dataset (weighted coefficient of determination - $wR^2$: 0.847; weighted mean absolute error - $wMAE$: 1.471 °C; weighted root-mean-square error - $wRMSE$: 1.836 °C). **f** External validation datasets ($wR^2$: 0.910; $wMAE$: 1.952 °C; $wRMSE$: 2.733 °C), the dashed line marks the 1:1 ratio. In (**a**–**d**) we show conditional regression plots; shaded areas represent the 95% confidence interval of the average estimates, points are partial residuals. Interactions were plotted using overlaying cross-sectional plots. The relationships between soil temperature and a continuous predictor are thus shown for several classes of the other predictor. When the classified predictor is continuous (i.e., panels **a** and **b**), we used its minimum, mean and maximum values. Source data are provided as a Source Data file.

confirmed a stronger temperature increase nearby glaciers (Supplementary Fig. 6); this effect was evident for all latitudinal bands, and more evident during warm seasons.

We estimated the potential for microclimatic buffering as the ratio between the current spatial variability and the temporal change experienced during the last 20 years. The sign of this relationship returns the direction of the temporal change (i.e. either temperature increase or decrease), while its absolute value measures the buffering potential (e.g. values of +2 or −2 indicate spatial variability twice the temporal variation, given increasing or decreasing temperatures, respectively). In the majority of cases, the fine-scale spatial variability of soil temperature within 250 m was larger than the temporal change, suggesting that it can play a relevant role for microclimatic buffering (Fig. 3c). Most buffering values (44.4% to 70.1%, depending on the latitudinal band and the distance from the glacier) had absolute values > 1 and ≤ 10 indicating that, in the last 20 years, spatial variability was one-to-ten times larger than the temporal temperature change. When considering all values ≤ −1 or >1 (i.e., looking at all the sites potentially guaranteeing buffering), percentages ranged between 52.9 to 80.2%.

The duration of the snow-free season estimated from satellite images increased between 2001–2005 and 2016–2020. At the global scale, the mean increase of season duration was 9.7 days (sd: 19.5 days;

Fig. 3d). Such an increase was larger nearby glaciers (16.2 ± 22.6 days) compared to areas located 3 km from a glacier (5.3 ± 15.7 days). In the Inter-tropical zone, the effect of distance from glaciers was particularly marked. Here, almost no change in season duration occurred in sites located more than 1 km away from a glacier, probably because in the tropics areas far from glaciers are almost constantly without snow.

## Discussion
Understanding the effects of climate change on high-elevation ecosystems is pivotal to predict the future of these threatened environments. Accurate microclimatic information is crucial to identify the conditions that are effectively experienced by living organisms, as changes in microclimate strongly influence local distribution and survival of individuals[24]. At the same time, the extreme environmental heterogeneity of mountain habitats, mainly generated by the altitudinal gradients, the complex topography and the variability of vegetation (from mature forests to grasslands, peatlands and tundra to bare soils) determines patchy microhabitats with a striking range of microclimatic conditions in relatively small areas, potentially buffering the severity of warming impacts on populations[35–37].

As expected, temporal changes in microclimate are tightly linked to climate trends at the regional or global scale, with macroclimate playing the major role in driving local temperatures. Our results

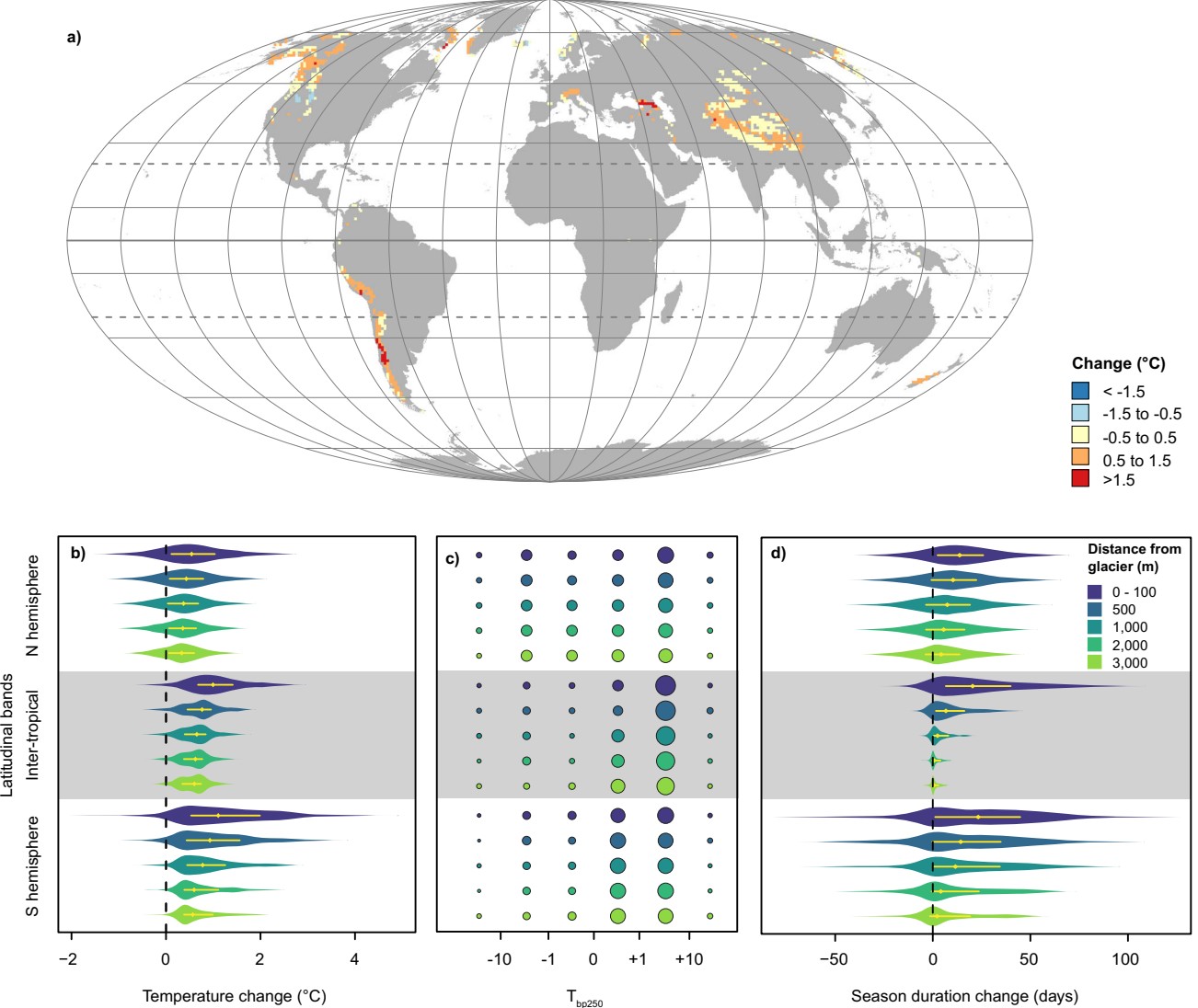

**Fig. 3 | Microclimate and season duration changes between 2016–2020 and 2001–2005. a** Per-cell average changes in soil temperature during the snow-free season; the dashed horizontal lines identify the Tropics, while the continuous tick one the Equator. Map projection: Mollweide (ESRI:54009); grid: 20*20 degrees; resolution: 100 km. A zoom on key areas is provided as Supplementary Fig. 3. **b** Violin plots summarizing the overall temperature trends for each distance class and latitudinal band (ΔT). **c** Percent distribution of the buffering potential of microclimate within a 250 m buffer ($T_{bp250}$) for each distance class and latitudinal band: absolute values greater ($T_{bp} > 1$) or smaller ($T_{bp} < -1$) than one indicate a microclimatic variability larger than temporal variation, potentially buffering large-scale climatic variations and allowing organisms to persist locally. The circle area is proportional to percentages. **d** Changes in the duration of the snow-free season, estimated from satellite-derived snow cover (NDSI). Violin plots summarize the overall trends for each class of distance from the glaciers and latitudinal band. In (**b**, **d**) yellow dots mark the median value for each series, while yellow lines the first and third quartiles. The Inter-tropical zone refers to the zone between the Tropics (23.44° S to 23.44° N); Northern and Southern hemispheres refer to areas north and south of either Tropic. Study area: 72° N – 60° S. Source data are provided as a Source Data file.

highlighted a generalized increase in soil temperature between 2001 and 2020 at all latitudes and distances from the glacier front, with the presence of clear seasonal trends. In both hemispheres, microclimate variation was stronger during warm seasons, while it was reduced during cold months. In the same period, we recorded an increase in the duration of the snow-free season, confirming at the global scale the results of regional analyses[31]. Both these changes were particularly evident in areas close to glaciers, where temperature is rising faster than expected on the basis of macroclimate (Supplementary Fig. 4). Such accelerated warming is probably linked to the shrinkage of ice with the consequent reduction of its cooling effect, and to the prolongation of the snow-free season[31]. The increasing number of days with snow-free terrains deeply influences soil temperature, mainly through the interaction with other drivers of temperature (Fig. 2). The absence of snow increases heat exchanges between air and soil, and

the lower albedo facilitates absorption of solar radiation, resulting in steeper relationships[25]. This amplifies the temperature increase along elevational gradients[44], and likely has major impacts on the whole ecosystem, such as an increase in vegetation productivity, and a change of biotic communities[31,33].

Owing to the mechanism of elevation-dependent warming, temperature increase is often faster in mountain areas than in surrounding lowlands[44,45]. Such accelerated warming poses strong challenges to mountain ecosystems, potentially leading to species altitudinal migrations, phenological changes and mismatches between different components of the ecosystem[46]. However, the impact of increasing soil temperatures and duration of snow-free season on local alpine biota may be partially counterbalanced by the spatial variability of microclimate conditions. For example, Maclean[47] recorded differences in temperature of almost 20 °C across a four-hectares study area,

corresponding to differences in average temperature across entire continents. When analysing the potential for microclimate buffering, we found conspicuous variations in soil temperature, with a mean value of about 1.2 °C within a narrow spatial range (250 m), i.e., one-to-ten times the recorded change during the last 20 years. Such spatial variability in microclimate conditions has key effects on local communities[33,48], and might allow individuals and even communities to withstand, at least temporarily, the effects of climate warming by modifying their distribution over relatively short distances. Nevertheless, fine-scale heterogeneity is probably not enough to buffer warming patterns expected to occur in the long term, as several climate change scenarios suggest that most mountain regions will experience a warming >4 °C by the end of the century[2]. Worryingly, microclimate is warming even faster than macroclimate in the areas closer to glaciers, with a particularly strong pattern in the tropical and Southern hemisphere mountains (Supplementary Fig. 4). In these regions, the probability of long-term persistence in microrefugia is particularly low, as climate change risks are amplified by the faster microclimate warming.

Our analysis focused on soil temperature, but water availability is a further parameter that should be taken into account when assessing microclimatic changes, as it influences multiple ecosystem traits such as biodiversity distribution[12,19,49], nitrogen mineralization[50], and the fluxes of carbon across the soil-vegetation-atmosphere interface[21]. Soil moisture can also influence local temperature, with moist soils having higher thermal inertia[51]. The combined effects of differences in soil moisture and temperature may result in microclimate patterns even more complex than those generated by temperature alone; still, the lack of high-resolution data on soil properties (e.g., texture, composition) hampered so far the modelling of soil water content at regional and global scale[40].

As far as we know, the dataset we assembled is the most complete collection of soil temperature recordings in proglacial areas. Although we tried to cover areas at different latitudes and with diverse climatic conditions, we acknowledge that data used to develop and validate our model are not truly global, thus may not be fully representative of the conditions in other mountain ranges with very different climatological characteristics. For instance, in some region glaciers are not currently retreating, owing to the combination of locally stationary temperatures, increasing precipitation and/or heavy debris cover (e.g., Karakoram [52]; in others they are retreating considerably, but following local dynamics (e.g., the southern Himalayas, conditioned by the Indian monsoon). A further extension of the dataset and its implementation with other initiatives (e.g., in ref. 41) might improve the global representation of the soil temperature dynamics, and possibly allow to better define the effect of some parameters (i.e., vegetation, permafrost) on soil temperature in glacier-related environments. Our analysis provides much better information on the microclimate of high-mountain environments than any currently available climate product (Supplementary Fig. 2), and enables an unprecedented view of the fine-scale spatial and temporal heterogeneity of climate change.

During the last decades, we boosted our understanding of the drivers of microclimate[24]. A huge amount of information is already available at the macro- and meso-scales for long time-series (e.g., monthly temperatures from TerraClimate), while others can be retrieved using remotely-sensed data. This information is fundamental to account for both interannual climate variability and long-term climatic trends. However, the effective and consistent modelling of microclimatic conditions over large areas still requires important efforts. Blending process-based models (e.g., in ref. 53) with data-driven empirical models (e.g., in ref. 21) and assimilating the data flow produced by remote sensing could be a first step to the construction of a "digital twin" of alpine ecosystems, as a specific contribution to global programs such as the EU initiative Destination Earth (https://digital-strategy.ec.europa.eu/en/policies/destination-earth). This, in turn, will be fundamental to model and understand relationships between mountain species and their environment, and to quantify their responses to climate change.

## Methods

### Temperature data and predictors

We focused our analyses on glacier forelands, i.e. the landscapes emerging after glacier retreat[3]. Data on soil temperature were collected at 175 sites from 26 glacier forelands located in the Svalbard archipelago (Norway; 2 forelands), European Alps (Italy, Austria, Switzerland and France; 21 forelands), and the Andes (Peru; 3 forelands), between 15th July 2011 and 24th August 2021 (Fig. 1). In each foreland, 1 to 16 devices (mean: 6.7; sd: 4.3; total: 175) were buried with no shielding at 5, 10 or 15 cm (mean: 9.2; sd: 3.0) below soil surface. The distance between devices within the same foreland ranged from 0.1 to 1798 m (mean: 149; sd: 223). Devices recorded temperatures for 33 to 763 days (mean: 501.9; sd: 191.7) with a recording frequency varying between 6 and 480 recordings/day (mean: 11.1; sd: 36.5). The total dataset was composed by 706,810 recordings; geographic attributes, device model, recording parameters and burying depth are reported in Supplementary Table 3. All months sampled for less than 90% of time were removed, and monthly averages were calculated based on the remaining data. The final dataset was composed of 2739 monthly average temperatures from 26 glacier forelands, ranging between August 2011 and July 2021. For each glacier foreland, the region of interest (ROI) was defined as the extent enclosing all the sampling stations, with a 750 m buffer as this enabled to include areas from the glacier tongue to downstream areas; a larger buffer (1500 m) was set up for Morteratsch and Dammagletscher forelands in order to include part of the glacier tongue within the ROI.

Macroclimate information for the sampled months and years was retrieved from the medium-resolution climate product TerraClimate[42] (resolution: 150 arcsec). TerraClimate was chosen as it provides time-series of monthly temperatures from 1958 to present, with yearly updates. This allowed relating soil temperature to macroclimate conditions in any given year and month between 2011 and 2021. Monthly mean temperatures were calculated as the midpoint between TerraClimate monthly minimum and maximum temperatures. To produce high-resolution macroclimatic surfaces while accounting for the adiabatic decrease of temperature, we downscaled monthly temperatures applying a fixed environmental lapse rate of −0.0065 °C/m[54]. To understand how sensitive the downscaled surface to the chosen lapse rate is, we also produced estimates using different lapse rates (−0.005 and −0.008 °C/m[55]). Estimates obtained using the different lapse rates showed very high pairwise correlation (minimum Pearson's $r > 0.98$), thus we decided to use the standard value of −0.0065 °C/m as suggested by Barry[54]. The high-resolution digital elevation data needed for downscaling macroclimate were retrieved from the ASTER GDEM v3 (resolution: 1 arcsec, i.e. approx. 30 m at the equator; latitudinal extent: 82° N to 83° S) via the NASA Earthdata interface (https://doi.org/10.5067/ASTER/ASTGTM.003; last accession on 24th March 2020). The high-resolution temperature surfaces obtained by downscaling TerraClimate simply represented the fine-scale variability of macroclimate related to altitudinal differences, rather than the effective topo- or micro-climates.

To account for the effect of absorbed solar radiation, we calculated the downward shortwave solar radiation following the approach implemented in the shortwavetopo function[39], with slope and aspect data retrieved from ASTER GDEM v3. The total shortwave radiation absorbed by a surface for a given hour, month and location is the sum of the diffuse and the direct components of radiation, taking into account the proportion of sky in view, solar altitude, azimuth, topographic shading and surface albedo. Monthly-averaged hourly estimates of total downward and net (i.e. albedo-free) radiation, as well as of surface albedo were retrieved from the ERA5-Land product[56]

(available at https://developers.google.com/earth-engine/datasets/catalog/ECMWF_ERA5_LAND_MONTHLY_BY_HOUR) using Google Earth Engine and the rgee R package[57]. The ERA5-Land product does not provide direct-diffuse separation (i.e. estimates refer to the sum of direct and diffuse components). To overcome this issue, we implemented the Yao2 separation method[58], as it showed the best performance in high-elevation and high-latitude (i.e., high-albedo) areas[59]. Specifically, we: i) used total downward radiation to estimate the hourly clearness index $K_t$; ii) used the Yao2 polynomial model to obtain the hourly diffuse solar radiation fraction $K_d$; iii) estimated hourly diffuse and direct net radiation (MJ m$^{-2}$ h$^{-1}$) by multiplying ERA5-Land net radiation for $K_d$ and (1-$K_d$), respectively. Coarse-grained direct and diffuse components of net radiation were then used to obtain high-resolution estimates of effective absorbed hourly shortwave radiation for the 15th of each month and year following the shortwavetopo approach[39]; we used the ERA5-Land forecast albedo product to represent the albedo from adjacent surfaces, contributing to the diffuse local component, and further partitioned the diffuse radiation in its isotropically-distributed, anisotropically-distributed and reflected back components. The above-detailed procedure was used to obtain 24 estimates of hourly shortwave radiation (i.e., we calculated one estimate for each hour of the 15th day of each month). The hourly estimates were finally summed up to obtain the monthly-averaged daily cumulative shortwave radiation (MJ m$^{-2}$ d$^{-1}$).

Snow cover (i.e. the presence of snow on the ground) causes local soil temperatures decoupled from regional climate[26], due to the insulating effect of the snowpack. To quantify this thermal effect on soil temperatures, for each month we assessed the proportion of days in which the device was under the snow. When a device is under the snow, it shows a very limited daily variation. We tested several values of diurnal range, assuming that a sensor was under snow when it showed a range below different threshold values (0.5, 1, 1.5, 2, 2.5, and 3 °C), and calculated the corresponding number of days with no snow on the ground for each of these thresholds. These estimates were compared with the ones obtained, for the same months and years, using the NDSI-derived coarse-grained MODIS Terra 500 m daily fractional snowcover[60] (available at https://developers.google.com/earth-engine/datasets/catalog/MODIS_006_MOD10A1). Fractional snow-cover was converted to snow occurrence using the conservative threshold of 40%[61], and the monthly frequency of snow-free days was estimated. Monthly estimates were based upon different number of images, mainly due to cloudiness and polar night (mean ± sd: 11.34 ± 5.37), consequently we sequentially discarded estimates based upon less than 1, 5, 10, 15, and 20 images, and tested the agreement between the number of days under the snow estimated from sensor and from the MODIS data. Differences in sample size between regions (Polar, Mid-Latitudes and Equatorial) and glaciers, as well as those in the monthly frequency of snow-free days within each glacier may inflate agreement scores. To account for these differences, we thus conservatively downweighted each observation, so that observations from each region sum up to 1, observation from each glacier within each region sum up to 1/G (G being the number of glaciers in the region) and observation within each glacier × region sum up to 1/M (M being the number of sfd categories for the glacier). The agreement between snow estimates from sensor and MODIS data was measured using i) the coefficient of determination from a weighted linear regression ($wR^2$), ii) the weighted mean absolute error ($wMAE$) and iii) the weighted root-mean-square error ($wRMSE$). We found a good match between sensor and MODIS estimates (maximum $wR^2 = 0.91$; minimum $wMAE = 5.13\%$; minimum $wRMSE = 13.96\%$). The proportion of snow-free days was thus calculated based on the threshold value of 1.5 °C, considering months with at least 15 images, as this combination provided the most robust estimates of snow-free days ($wR^2 = 0.91$; $wMAE = 6.58\%$; $wRMSE = 13.96\%$). In principle, this approach might bias the model toward less cloudy areas, as they generally have a larger

number of images per month. Nevertheless, we had to limit analyses to months with a minimum number of images to avoid inaccurate sfd estimates.

The presence of nearby glaciers represents a further driver of local temperature, for instance because of katabatic winds. To account for this cooling effect, we measured the distance between each sampling station and the glacier front, under the assumption of decreasing cooling effects with distance[62]. For each glacier, the most recent outline was retrieved from Marta et al.[63], checked against glacier position between 2015 and 2019 using USGS Landsat 8 imagery (available at https://developers.google.com/earth-engine/datasets/catalog/LANDSAT_LC08_C01_T1_TOA) and eventually updated to include other nearby glaciers or more recent outlines. Outlines were transformed to polygons, rasterized, and distance maps at 30 m resolution were calculated using the function gridDistance from the raster R package[64]. Three categorical variables were calculated to account for the effect of (i) tree cover, (ii) permafrost occurrence and (iii) differences in the depth of logger burying. Tree shading can decrease soil temperature, mainly reducing absorbed radiation, thus we used the 30 m resolution Hansen Global Forest Change v1.8[65] (available at https://developers.google.com/earth-engine/datasets/catalog/UMD_hansen_global_forest_change_2020_v1_8) to assess if a logger was in a tree-covered area or not. Original cover data, expressed as percentage of per-cell canopy closure were converted to tree occurrence for all the cells with > 0 tree cover. Coarse-grained data on permafrost extent (30 arcsec) were retrieved from Gruber[66] and dis-aggregated at the 30 m resolution. To convert permafrost extent (expressed as the proportion of a cell that is underlain by permafrost) to occurrence of continuous/extensive discontinuous permafrost we used the conservative 0.9 threshold, i.e. we considered permafrost present when its probability of occurrence was ≥ 0.9. Lack of high-resolution, global data on vegetation cover and height hampered the introduction of modelling terms accounting for both longwave radiation and soil shading by herbs and mosses.

## Model calibration

Monthly-averaged observed soil temperature (soilT) was modelled using linear mixed models (LMM). As independent variables we used downscaled macroclimate (mT), monthly-averaged daily cumulative shortwave solar radiation (rad), monthly frequency of snow-free days (sfd), distance from glacier forefront (dg), tree cover (tc), permafrost occurrence (pf) and depth of burying (d). Interactive terms were added to account for the effects of a varying frequency of snow-free days (sfd) on mT, rad, and dg, as well as for the potential effects of tc on rad and of d on sfd. To include geographical factors not explicitly accounted for by the selected set of predictors, we additionally incorporated a random intercept on glacier (1|gl). Consequently, the full model takes the form:

$$soilT \sim mT + rad + sfd + dg + tc + pf + d + sfd : mT \\ + sfd : rad + sfd : dg + sfd : d + tc : rad + 1|gl \tag{1}$$

Winter snowpack decouples air and soil temperatures causing no relationship between soilT and several predictors (e.g. air temperature, solar radiation) during seasons with snow. To remove the effects of this decoupling, while reconstructing soilT during the snow-free season, we i) classified sfd in 10 intervals between 0 and 100%; ii) run iteratively the model retaining only records with sfd > 10%, 20%,…, and iii) plotted fitted values vs residuals to evaluate the residual structure at each step. With sfd > 20% the effect of decoupling was almost completely erased. Consequently, all the months with sfd ≤ 20% were discarded, and the resulting dataset included 1,516 monthly average temperatures from 26 glacier forelands. Before running the final model, dg was square-root transformed to linearize the relationship with the response variable and all the continuous predictors were

scaled to zero mean and unit variance. Linear mixed models were run using the lme4[67] and lmerTest[68] R packages. Model residuals approximated a normal distribution (Shapiro-Wilk test; W = 0.994), and the variance inflation factor was low (GVIFadj$_{max}$ = 2.79, including interaction terms), indicating that multicollinearity did not pose major issues. Model performances were evaluated using Nakagawa and Schielzeth[69] $R^2$, as implemented in MuMIn R package[70]. The amount of variance explained by single model terms was quantified by calculating the semi-partial $R^2$ using the partR2 R package[71], with 1000 bootstrap replicates. partR2 iteratively removes predictors and compares the change in variance of the linear predictor to the variance explained by the full model; higher the difference between the two values, higher the amount of variance explained uniquely by a given predictor. We followed Thuiller et al.[72] to account for the overall effect of single predictors (i.e. considering their joint contribution to both additive and interactive terms). Each predictor was randomized 1000 times, and the predictions obtained using original and randomized datasets were compared via the Pearson's correlation coefficient (r). Strong correlations indicate that randomizations had little effect on model performances; for each permutation, variable importance was finally expressed as 1-r.

## Model validation

The performance of the model was assessed using both internal (with the data used to build the model) and external validation (with fully independent data). For internal validation, we used a leave-one-out approach. We iteratively run the full model, retaining all glaciers except one, and the estimated fixed coefficients were stored. We then used the average coefficient to predict expected temperature without accounting for glacier identity. The agreement between observed and predicted temperatures was measured using $wR^2$, $wMAE$ and $wRMSE$, following the same weighting scheme applied during the calculation of monthly frequency of snow-free days.

External validation, using data that are fully independent from the ones used to calibrate the model, is pivotal to assess the actual transferability of models, and thus its applicability at the global scale. External validation was based on an updated version of the SoilTemp database[26] (available at https://doi.org/10.5281/zenodo.4558663). To obtain data comparable to those used during model training, we selected sites with depth of burying 5–15 cm and distance from glaciers (measured using the GLIMS glacier outlines[73]) ≤ 3000 m; the validation dataset was further cleansed removing all stations on water bodies / rivers or human infrastructures (i.e. roads). The resulting dataset was composed of 6472 monthly recordings from 170 stations, distributed in Europe (Alps, Pyrenees and Scandinavia) and Asia (Himalayas, with stations in India and Nepal). For each record, predictions of soil temperature in a given month and year were obtained on the basis of the mean coefficients from the leave-one-out analysis, following the procedure described for the global projection of soil temperature (see below). Depths were associated to the nearest group (5, 10, or 15 cm). We discarded records with estimated snow-free days based upon < 15 images, and months with sfd ≤ 20%. This reduced the dataset to 1518 monthly recordings in 161 stations. To obtain spatially unbiased goodness-of-fit statistics, we implemented a weighting scheme similar to that used for the training dataset. Given the lack of the "glacier" level, we grouped all devices closer than 1000 m each other in one cluster. All observations were then downweighted, so that observation from the same region (southern Europe, Scandinavia or Himalayas) sum up to 1, observation from the same geographic cluster within each region sum up to 1/C (C being the number of clusters in the region), and observation within each cluster × region sum up to 1/M (M being the number of sfd categories for the cluster).

To confirm that temperatures estimated by our model approximate the actual temperature better than other already available products, we also compared observed temperatures of both the training and validation datasets to the ones predicted by our model and the time-series of TerraClimate[42] (resolution: 150 arcsec) and CHELSA[43] (resolution: 30 arcsec). For each observation, we extracted climate data for the corresponding year and month, after excluding the observations from 2020 and 2021 (the CHELSA time-series being limited to 1979–2019), and calculated $wR^2$, $wMAE$ and $wRMSE$.

Our model focused on mean monthly temperature, but other parameters (e.g., minimum or maximum temperature) can be important for organisms. We thus checked the correlation between mean temperatures of both training and validation datasets, and minimum and maximum monthly temperatures. Following Lembrechts et al.[26], we calculated minimum and maximum monthly temperature as the 5% and 95% quantiles of monthly values. Monthly mean temperature was highly correlated to temperature extremes (Pearson's r > 0.91 for both minimum and maximum temperature). This strong relationship between temperature average and extremes is possibly due to the rather scarce and homogeneous soil cover, which is mostly occupied by sparse vegetation and high-elevation tundra and has a different behaviour from what is observed in forests[34]. This suggests that mean temperature provides a good representation of the overall pattern within each month.

## Global projection of soil temperature

Obtaining high-resolution estimates of soil temperature in glacier forelands, at the global scale and in several periods, allows estimating soil microclimate variability and temporal variation, measuring the impacts of climate change on microclimate and the potential for microclimate buffering. The aim of this analysis was to assess the variation of microclimate during the last decades, thus we compared microclimate between the periods 2001–2005 and 2016–2020. We used the mean coefficients obtained from the leave-one-out analysis to generate predictions of soil temperature at the global scale, using Google Earth Engine and the rgee R package[57]. Due to data availability, the analysis was spatially constrained between 60° S and 72° N. We focused on proglacial landscapes, thus we limited projections to within 3 km from glacier outlines. It is worth noting that some differences exist between the training and global projection for: the digital elevation products, the approach to glacier outline identification, the definition of the monthly frequency of snow-free days and the calculation of shortwave solar radiation.

Digital elevation data are needed for downscaling macroclimate and for calculating the daily cumulative shortwave solar radiation. For the global projection, we used a coarser resolution (90 m instead of 30 m) to limit computation time. From 60° S to 60° N we used the 90 m resolution composite of Shuttle Radar Topographic Mission v4[74] (available at https://developers.google.com/earth-engine/datasets/catalog/CGIAR_SRTM90_V4), while from 60° to 72° N we used the Global Multi-resolution Terrain Elevation Data 2010[75] (available at https://developers.google.com/earth-engine/datasets/catalog/USGS_GMTED2010), given that the SRTM model was not available above 60° N. Monthly mean temperature was calculated from TerraClimate (https://developers.google.com/earth-engine/datasets/catalog/IDAHO_EPSCOR_TERRACLIMATE). Monthly mean temperatures were obtained by averaging monthly minimum and maximum values across each five-year period, and downscaled to 90 m resolution following the same approach used for the calibration data.

Information on shortwave solar radiation was obtained as detailed in the previous section, but the shortwavetopo function was re-coded to be launched directly in GEE via the rgee interface (see Supplementary Software 1). For each cell, daily cumulative solar radiation was estimated for the 15th day of each month in the years 2003 (for 2001–2005) and 2018 for (2016–2020), and considered representative of the whole month and period. The monthly frequency of snow-free days was calculated using the NDSI-derived daily fractional snowcover as detailed in the previous section. For each month, we estimated the

percent snow occurrence using monthly values averaged over each five-year period and bilinearly-interpolated at the 90 m resolution. All cells with sfd values ≤ 20% or with sfd values based upon less than 15 images over the five-years period were excluded. To account for distance from the glacier, we used glacier outlines of the GLIMS dataset[73] (available at https://developers.google.com/earth-engine/datasets/catalog/GLIMS_current). Glaciers may have been retreating between 2001–2005 and 2016–2020; consequently, for each period and glacier ("glac_id" field), we selected the outline with the temporally closer source image ("src_date" field), and calculated distances according to those positions. Permafrost extent was uploaded in GEE, and bilinearly-interpolated at the 90 m resolution, while tree cover was aggregated at the same 90 m resolution. In projections, we estimated soil temperature at 5 cm depth ($d = 5$). Despite the methodological differences, shortwave radiation and temperatures estimated with the global model (90 m resolution) showed excellent agreement with the ones at the 30 m resolution (Pearson's $r = 0.93$ and $0.97$, respectively).

Maps of predicted soil temperatures at 2001–2005 and 2016–2020 pose some problems in handling and obtaining summary statistics at the global scale (2 periods × 12 months × $6.628 \times 10^{10}$ pixels; approximate size ≈ 4.7 TB). To overcome these limitations and obtain a spatially unbiased representation of microclimate variability and variation, instead of using all the cells we subsampled them using a stratified grid sampling by i) building a regular 50 × 50 km grid (Mollweide projection; ESRI:54009); ii) retaining all the grid cells containing glaciers or within 3 km from glacier outlines (2,604 cells), and iii) defining five classes of distance from the most recent glacier outline (0–100, 400–600, 900–1100, 1900–2100 and 2900–3100 m). The most recent glacier outline was the one used for calculating the distances for the 2016–2020 projection. Within each cell, we randomly sampled 10 points, two for each distance class. The resulting dataset was composed of 26,040 points, each associated with 12 × 2 (2001–2005 and 2016–2020) measures of monthly soil temperature. After removing points with missing temperature estimates for all the months in one or both periods, and cells with < 10 points (e.g. because some points were in the sea), the final dataset was composed of 19,440 records from 1944 cells. For this set of points, we extracted the monthly average temperature for the two periods. Based on temperature data, we calculated both annual and seasonal (Dec–Feb; Mar–May; Jun–Aug and Sep–Nov) microclimate variation (ΔT) between the two periods ($\Delta T = T_{2016–2020} - T_{2001–2005}$). For the same set of points, we also extracted the annual duration of the snow-free season for the two periods. We measured the snow-free season as the total number of days with no snow on the ground (i.e., with fractional snowcover < 40%[61]) during the whole year, averaged over each of the two periods.

Short-distance movement of individuals might allow buffering the severity of warming impacts on populations, if suitable climatic conditions occur nearby[35]. To understand the potential for microclimate buffering of proglacial environments, we compared the recorded microclimate variation between 2001–2005 and 2016–2020 (ΔT) to the spatial variability of soil temperatures. The spatial variability of microclimate was calculated as the 80% inter-percentile range within a 250 m buffer ($T_{var}$). Due to computing limitations, the analysis only considered the average microclimate (mean annual temperature) of 2016–2020. Microclimate buffering potential ($T_{bp}$) was calculated as: $T_{bp} = T_{var} / \Delta T$. This formula allows measuring both the direction of the change and the buffering potential, as it retains the sign from ΔT (e.g. positive values indicate temperature increase), but returns (absolute) values > 1 ($|T_{bp}| > 1$) when the spatial microclimate variability is larger than the temporal microclimate variation.

Extrapolation beyond the conditions experienced during training and validation datasets can determine limited transferability of model predictions[76,77]. In order to assess extrapolation issues, we tested whether independent variables used for global projections have values falling outside the range observed in the training and validation datasets[76]. The majority of independent variables (distance from glacier, permafrost and tree occurrence, percent of snow-free days) did not show extrapolation issues, as their values were within the range experienced during training. Some extrapolation occurred for downscaled macroclimatic temperature and shortwave solar radiation, still extrapolation levels were very limited. For temperature, just 0.75% of values were outside the range, while for solar radiation just 2.35% of values were outside the range, suggesting no major transferability issues[76].

## Data availability
Soil temperature data used in this study to train and validate the model have been deposited in a FigShare repository at https://doi.org/10.6084/m9.figshare.23736966[78]. The data generated in this study and used to build the Figures are provided in the Source Data file. Source data are provided with this paper.

## Code availability
Main code to run the global projection is made available as Supplementary Software 1. The code used to run the model, the internal and external validation and to produce Fig. 2, Supplementary Fig. 1 and Supplementary Tables 1 and 2 has been deposited in a FigShare repository at https://doi.org/10.6084/m9.figshare.23736966[78].

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

## Acknowledgements

S.M., M.C., M.G., R.A., R.S.A., F.G., F.P., W.T., and G.F.F. were funded by the European Research Council under the European Community's Horizon 2020 Programme, grant agreement no. 772284 ('IceCommunities-Reconstructing community dynamics and ecosystem functioning after glacial retreat'). AZ was funded by the National Geographic Society (grant no. EC-58123R-19) and the United States National Science Foundation under the Human-Environment and Geographical Sciences Program (grant no. 2105826). This research was also funded by Biodiversa+, the European Biodiversity Partnership under the 2021-2022 BiodivProtect joint call for research proposals, co-funded by the European Commission (grant agreement no. 101052342 'PrioritIce-Vanishing habitats: conservation priorities for glacier-related biodiversity threatened by climate change') and with the funding organisations MUR and ANR.

## Author contributions

S.M. and G.F.F. conceived and designed the study. S.M., A.Z., M.C., M.G., R.A., R.S.A., F.G., F.P., and G.F.F. collected and validated field data. S.M. wrote the codes, conducted the statistical analyses, and produced the illustrations with major inputs from G.F.F., A.P. and W.T. S.M., A.Z., M.C., M.G., R.A., R.S.A., F.G., F.P., W.T., A.P. and G.F.F. contributed to writing the article.

## Competing interests

The authors declare no competing interests.
