## [Peer Review File · Nature Communications]

Heterogeneous changes of soil microclimate in high mountains and glacier forelandsREVIEWER COMMENTS

Reviewer #1 (Remarks to the Author):

This is a very nice paper that produces a global dataset of microclimate change from 2001 to 2020 to estimate whether microclimate heterogeneity might buffer the severity of warming impacts on organisms. Most of my comments relate to the modelling component of the manuscript, which lies most within my areas of expertise.

In general, I applaud the use of a hybrid correlative-mechanistic approach to do the microclimate modelling, as it allows high-resolution datasets to be generated over larger extents and durations, and doesn't suffer from some of the same drawbacks of purely statistical approaches. There are, however, several improvements that could have been made to the models to better account for mechanisms underpinning microclimate. The only one I insist on is that radiation is correctly calculated and defined. The problem here is that it seems that the solar index was applied to total (shortwave) radiation rather than direct radiation. The authors also don't explicitly mention longwave radiation and tend to use the generic term radiation when I think they mean downward shortwave radiation.

Downward shortwave radiation intercepted by a surface is more correctly calculated as:

Direct beam radiation x solar index + diffuse radiation x sky-view.

Total incoming radiation can be partitioned into its direct and diffuse components and sky-view factors calculated using function in the microclima R package (Maclean et al 2019 *Methods Ecol Evol*, 10: 280-90) which is also a better reference for the solarindex function than the citation given.

Doing it this way would omit the need to include cloud cover as a term in the statistical model, thereby simplifying it significantly.

It would be more appropriate also to calculate absorbed rather than intercepted radiation by incorporating surface albedo (absorbed = (1-albedo) x intercepted). There are global albedo datasets, but a simple framework would be to assume a surface albedo of ~0.15 without snow cover and ~0.9 with snow cover. This could further simplify the statistical model by omitting the need to include interactions between snow and radiation and cloud cover and snow.

Note also that the main reasons that tree cover have an effect on ground temperatures is because it intercepts shortwave radiation, so it should really be included as an interaction term with radiation (as opposed to an additive term). A simple way to handle this pseudo-mechanistically would be to apply a correction to shortwave radiation prior to performing the terrain correction. Shortwave radiation below canopy is given approximately by (1-fractional canopy cover) * shortwave radiation above canopy. There is a secondary effect of trees on surface roughness, which in interaction with wind speed controls the slope of the relationship between radiation temperature offsets and net radiation.

Overall, by going slightly further down the mechanistic route of modelling microclimate, the authors could actually significantly simplify their statistical model without adding much computational penalty, with the added advantage of giving their models greater physical meaning. The route to do this is to recognise, following Bennie et al (2008 Ecol Model 216: 47-59 and the Maclean ref given above that

$$T_{\text{micro}} = T_{\text{meso}} + d \cdot (R_{\text{net}} - L - G)$$

where T_{micro} and T_{meso} are microclimate and mesoclimate temperature respectively, R_{net} is net radiation (including the longwave component, which would need to be handled sensibly), L is latent heat and G , is the rate of heat storage in the ground and d a measure of the degree of thermal coupling between the surface and the air, a function of wind speed and surface roughness. By mesoclimate I mean macroclimate corrected for elevation and e.g. distance to glacier. G and L can be considered small and proportional to net radiation so the temperature offset from macroclimate (ignoring mesoclimate effects) is a linear function of net radiation, the slope of which is determined by d . Since d scales linearly with $\log(\text{wind speed}) \times \text{vegetation height}$, there is a direct route to calculating microclimate, though one would still need to account for some of the mesoclimate effects and snow cover. There are slight issues in applying this approach to monthly data, as is generally used when modelling using sub-daily time-steps, but since most of the relationships are linear, issues associated with Jensen's inequality only manifest in the calculation of d using $\log(\text{wind speed})$. However, I suspect this issue is quite minor and a nice model could be derived using this approach. Nevertheless, I leave it up to the authors to decide whether following this approach might simplify their model and make it more robust.

Aside from this set of comments about the modelling, most of my comments are quite minor:

Justification for calculating mean as opposed to say monthly maximum and minimum temperature is not really provided, and I wonder if by not doing this they are missing some interesting patterns. Certainly one might expect spatial heterogeneity in maximum temperature to be more pronounced than in mean temperature and the extremes may also be more biotically important.

Lines 48-50. Potentially slightly misleading as this presumably is largely to do range overlap meaning that any 10% of the planet would host >10% of species (rather than it being the case that mountains are particularly diverse).

Lines 63-64. Consider replacing 'topography' with 'terrain'

Lines 67-68. More pronounced just above rather than below. But that doesn't negate the point you are making.

Lines 68-70. Worth mentioning elevation lapse rates here?

Lines 81-83. Not quite clear whether you are talking about spatial or temporal buffering here, but

actually temporally near-ground temperature has greater variance than the shade temperatures measured by weather stations for reasons the authors themselves outline in lines 68-70. Not true under snow cover though.

Lines 489-506. It would be useful to know which devices were used to obtain temperature measurements as different devices have different degrees of accuracy (though wildly inaccurate results is much more of an issue when measuring air temperature in sunshine). It would also be helpful to be reassured that 'devices' were located in manner that captures the range and appropriately stratifies determinants of microclimate (though I recognise that there are significant logistical challenges here).

I hope my comments are useful

Ilya Maclean (I have a policy of signing my reviews where they make reference to my own work)

Reviewer #2 (Remarks to the Author):

This research uses an extensive network of temperature sensors which have measured soil temperatures in the areas surrounding glaciers to attempt to quantify the large microclimate variability in such regions and evaluate the consequences for the foreland ecosystems in the context of climate change. The complex topography, coupled with high elevation and variable snow cover, means exposure to (often intense) solar radiation is extremely variable, which can create strong instantaneous temperature gradients and high amounts of spatial variability. A model is developed to predict monthly soil temperatures based on such local factors and then applied globally to model changes in microclimate variability over the last 20 years. The main findings of the paper are that a) changes in microclimate have been greater nearer glaciers than further away and b) that local temperature variability in the vicinity of glaciers is extremely large and outweighs possible climate change forcing, at least at the current time, meaning that ecosystems may have a buffer to withstand some of the more detrimental effects of rapid global warming.

In one sense, the paper tells us something we already know, since the idea of refugia (created by strong microclimates) is not new. However, the dataset I think is unique and the modelling methods are quite comprehensive and the model fairly successful, at least in statistical terms. I am not an expert in all of the statistical methods, but the models appear robust and methods are taken to assess their reliability. The paper is well written and referenced and there is a clear understanding of the relevant literature which sets the work in an appropriate context.

There are some limitations. Despite being applied globally, the model is not really global because of the limited and uneven concentration of data which goes into it. Although field data from 26 glacier forelands are included, 21 of the sites are from the Alps, and only 3 from Peru (tropical) and 2 from Svalbard (arctic). Thus, the model is dominated by mid-latitude maritime conditions. Taking out one glacier at a time to validate the model still means all runs are still Alpine dominated and I would therefore not expect the model coefficients to vary much (when using 26 rather than 27 locations) so the conclusion that the model is robust based on this alone is making a bit of a leap of faith.

There is little analysis to convince readers that it works just as well in the “minority” locations as it may do in Alpine locations. Obviously, the best solution is more comprehensive data, but I realise that it probably is not available. At the very least therefore it would be important to look at specifically how the non-Alpine forelands are predicted by the Alpine dominated model? i.e. does it work in the tropics or in the Arctic just as well as on the other glaciers?

Beyond the “leave one out” approach, it might be helpful to develop models separately for the three latitudes to see if they are different or similar (I realise sample sizes are uneven but that is unavoidable). Similar coefficients for the tropical glaciers for example will increase confidence that the model is indeed globally applicable. Different coefficients will cast light on variable microclimate controls. The mid-latitude Alps are also fairly maritime, and continental/arid glaciers such as those on parts of the Tibetan plateau, or at extremely high elevations in the tropics (e.g. Kilimanjaro) are missing from the model. It would be great if it could be validated on such a contrasting example. So in summary, at the moment the paper requires some more critical analysis to show that when the model is extrapolated globally into regions with quite different climatic regimes, that it is reliable.

There is also some limitation temporally. The data set has 2203 monthly readings from 175 sites which means an average of 12.6 months from each site. I realise these are not consecutive because they represent snow free periods, but even so this is not many years at each site, and they are for different years at each location. Assuming a typical snow free season is 4-6 months then we have about 2-3 summer seasons on average at each site. So variability between seasons and years (which could also be different between each location) must contribute to the results and success of the model. I understand all this is difficult to quantify, but it is important to discuss.

I also have some minor comments/concerns (see line numbers)

Line 67: Microclimate variance can be extreme above and at the surface, but actually just below a lot of this variability is reduced since the soil temperature shows a damped response. So it is a bit misleading to group near and below surface together (they are very different).

Line 128: Cloud cover has a greater effect when there are fewer snow free days (i.e. more snow). This does not make physical sense (surely more snow means a weakened response to radiation). It could be because solar radiation is typically strongest in late spring/early summer when snow is still melting (assuming northern hemisphere here)? So the relationship at the monthly scale is due to this seasonal association and not any physics? Perhaps this just needs a bit more clarification.

Figure 2: The axes have strange irregular units.... i.e. -0.69, 2.65, 6 etc for soil temperature... they need to have regular intervals. Some axes do not have units at all (i.e. solar radiation). Also why are individual values given for high, low and medium snow free days and cloud cover (surely they should be ranges?)

L 145: Downscaled macroclimate: it is not clear what this is. I think this is just TerraClimate altered with a standard lapse rate? I suppose the TerraClimate grid cell elevation may be very different from the foreland/datalogger elevation and so that is why a correction is required? If so I think this needs to be explained. The other question is how reliable this lapse rate correction is. As we all know, lapse rates are very variable in space and time, and I would think this might be a major source of uncertainty – but there is no sensitivity analysis of the decision to use a fixed lapse rate.

Line 250: The lack of change in snow cover season length as you move away from tropical glaciers is interesting and yes does imply that it might be due to lack of snow cover (so lack of snow cover change). However, in the tropics there is limited seasonality (at least due to temperature) and in many cases

snow falls intermittently throughout the year (mostly in wet seasons) only then to disappear in days/weeks so there is no well-defined snow season in the first place. I wonder whether it is not clear what you are measuring.

Line 286: Change to "temperature increase is often faster" (not always I would argue).... On average positive EDW has been shown to exist, but there are places at high elevation which are not warming faster.

Line 311: Change to "Although we tried...."

Line 316: Change to "in others they are retreating considerably"

Line 319: Change to "Our analysis provides...."

Extended Data Tables 1 and 2: This should be largely understandable from the caption alone and so a bit more information about how the coefficients were determined would be useful. In Table 1, what are d(10) and d(15) – dummy variables?

Line 527: It is strange that cloud cover is taken from a different time period (2000-2014). Why? Would it not be better to use the same period as the soil temp data?

Line 530 ff: Snow cover – I know you can get this from the logger temperature amplitude but would it not be better to use MODIS NDSI anyway (since this is transferrable) in the original model. At the moment I think you use it to determine the threshold amplitude, but then define snow days using the temperature and not NDSI. This might be more accurate for this model, but eventually in the global model you have to use NDSI anyway (since real temperatures are unavailable) so this just creates differences between the model which is tested and calibrated and the one used for global extrapolation.

Line 558/559 and 661: Tree cover is not really binary is it? Particularly in the glacier foreland where isolated trees might be common. The text implies it was yes or no, but surely a percentage value would have been more appropriate. There is lack of detail on this. What was the threshold?

Permafrost: Line 561 ff: What depth are we talking about when we define permafrost? I suspect this is at a much deeper location than the temperature loggers, and if so, I am not sure what the relevance is, again of a binary variable.

Line 669: Change to "instead of"

Line 671: How many points in each cell (on average?). I don't understand later how there can be fewer than 9 points in a 50 by 50 km cell?

To summarise I think there are some valuable insights here, but the main weakness is the uneven distribution of data which goes into the model (basically the Alps dominate the model), and the assumption that it can be clearly extrapolated globally with limited error. More analyses to support this contention and a broader discussion of applicability to different climate regimes is required. Other than this I think the paper provides potentially valuable insights and I would encourage the authors to make appropriate revisions.

Reviewer #3 (Remarks to the Author):

In this paper, the authors model the soil temperature in glacier forelands using data from three mountain climatically-distinct mountain regions across the globe. They disentangle the drivers of these

soil temperatures and use their model to assess soil temperature changes globally over the last two decades. While I applaud this project – quantifying microclimate change is urgently needed, especially in vulnerable environments like mountains – I have a few comments that need clarification.

Most importantly, the global coverage of points for model calibration is limited to three mountain regions, and I think that's a bit limited. The authors acknowledge this in passing, but don't take further action in this regard. What comes to mind as things that I find missing most are: 1) out-of-sample model validation: would your model be able to predict temperatures accurately in another mountain region? And 2) a masking of areas outside the model parameter space (with this I mean: points that have environmental (and especially climatic) conditions not within the range of any of your data points should be masked from the global assessment, as here it becomes increasingly hard to trust model extrapolations.

For the first point, I don't think the leave-one-out method is sufficient, as you always keep all three your mountain regions in the model. You might really have to look for another dataset from another mountain region (or test the predictive power of the model if you for example remove the whole of Svalbard. This would obviously be drastic, but the same could be set of using these three mountain regions to predict what happens in the Himalayas without verification. For the second point, at least checking the environmental space of your observations and the rest of the globe, so a reader can assess if this extrapolation might be problematic, would be a good start.

Specific comments:

L37: make clear that this 1-to-10 times is about the warming that so far has happened, and not about 'climate warming' in general – a reader who only reads the abstract will otherwise miss this nuance, I think. The climate warming that happened so far is significantly smaller than the climate warming to expect in the near future, even in the best case scenarios. You mention this in the discussion, but in the abstract the nuance is easy to overlook.

L64: perhaps say 'partially decoupled'?

L84: given that you just defined 'buffering' in the microclimate context above, I would use another term here. 'Limit'?

L84: not sure this concept only works for moving individuals, I think this is often (more often?) considered for populations

L90-104: I would here already explain how glacier retreat/advance is incorporated in your temporal analysis, as the fact that you focus on glacier foreland microclimate change makes that a key parameter and its not explained here that this is incorporated.

L103-104: how did you make this comparison with other high-elevation environments? Perhaps specify the used methodology here.

L120: is this also already downscaled macroclimate? The first mention of downscaling comes much lower. Perhaps also say that this is lapse rate-based downscaling and not including any other meso- or microclimate parameter.

Fig. 2g: one would expect a significant interaction between depth of burial and seasonality, with deeper sensors showing higher temperatures in autumn and towards winter, and shallower sensors showing higher temperatures in spring and summer, no? Yet this was not tested, I understand.

Fig. 2: the strong interactive effects of snow free days makes me ponder. As you used some kind of truncation approach of snow free days, how much of that is still just the buffering effect of snow?

L164: is the MAE in °C?

L175: I'm not entirely sure it's meaningful that your model – calibrated on your in-situ measurements – is more strongly correlated with those in-situ measurements than any other product not related to your in-situ measurements. This validation feels rather circular to me. This would make more sense to me, personally, if done out-of-sample on an independent dataset.

Extended data figure 2: here, to me, having the Root Mean Square Error – which gives higher value to extreme outliers – would be more logical to show the difference, no?

Fig. 3: what to me seems more important than absolute changes in temperatures is how much these temperature changes differ from macroclimatic observations of climate change. Trends will be the same, but it allows a reader to put things in a better perspective.

Fig. 3a: would it make sense to zoom in on one or a few mountain region(s) as an example?

Fig. 3a: it's surprising to me that Svalbard is excluded from your predictions, even though it's one of the three mountain regions where you have data, would it make sense to model for the whole globe and then use a mask to exclude areas out of the range of calibration?

Fig. 3c: somehow lacks resolution. Would some kind of density plots not work better?

Fig. 3d: based on the main text (L246-248) this seems satellite-based, is that correct?

L194-204: the high amount of numbers in this paragraph make it difficult to process them

Extended Data Fig. 3: why is the spatial extent of the maps different in each season? I'm assuming the answer is on line 655, but that would best be specified here.

L271: in your analysis, these aren't really trends, but spatial patterns, no? The temporal trends are just a space-for-time extrapolation that is not tested

L302: the discussion of soil moisture comes a bit out of the blue, as this was no parameter in your analyses.

L309: abrupt switch to lack of global coverage from the soil moisture story. New paragraph, perhaps?

L329: I feel like you should explain the concept of the 'digital twin', why it is important. Ecosystems to me have so many more aspects than microclimate, so this might be a big leap to a digital twin of the ecosystem, here. It could be a 'digital twin' of the abiotic conditions or something, perhaps? Or I'm misunderstanding the concept of the digital twin...

L489: how is a glacier foreland defined here?

L526: cloud cover has a coarse spatial resolution – especially for mountains, is it worth discussing this in the discussion as a potential reason for flaws?

L566: was month not included as a factor in your models? Shouldn't it be, at least as a random effect to correct for unaccounted seasonal effects? Otherwise, it could be pseudoreplication as well. The latter is reflected in Fig. 1, where you are actually depicting sensors x months rather than sensors.

L581-586: isn't the effect you are trying to remove here not also exactly the one driving the interaction with snow free days in your model?

L673: why 3100 meter? I thought you limited the analysis to 3000 m?

I encountered a few language errors, especially in the introduction and methods, that merit a dedicated verification.

Finally, as a little extra: please consider submitting your data to the SoilTemp-database (www.soiltempproject.com) and indeed consider using this database for future analyses, as you suggest yourselves on line 318. This could greatly enhance global coverage of your – and our – analyses. Given

the existence of the SoilTemp database, however, I would remove your claim of being the most extensive dataset on high-elevation soil temperatures (L310). This might be true for the particular case of glacier forelands, but most certainly not for high-elevation areas, regardless of your definition of 'high-elevation'. This made me realize: perhaps the 'high mountains AND glacier forelands' in the title is a bit misleading?

I hope these comments are helpful,

Kind regards,

Jonas Lembrechts

DETAILED RESPONSE TO THE REVIEWERS' COMMENTS

RESPONSE TO COMMENTS OF REVIEWER #1

This is a very nice paper that produces a global dataset of microclimate change from 2001 to 2020 to estimate whether microclimate heterogeneity might buffer the severity of warming impacts on organisms. Most of my comments relate to the modelling component of the manuscript, which lies most within my areas of expertise.

In general, I applaud the use of a hybrid correlative-mechanistic approach to do the microclimate modelling, as it allows high-resolution datasets to be generated over larger extents and durations, and doesn't suffer from some of the same drawbacks of purely statistical approaches. There are, however, several improvements that could have been made to the models to better account for mechanisms underpinning microclimate. The only one I insist on is that radiation is correctly calculated and defined. The problem here is that it seems that the solar index was applied to total (shortwave) radiation rather than direct radiation. The authors also don't explicitly mention longwave radiation and tend to use the generic term radiation when I think they mean downward shortwave radiation.

Downward shortwave radiation intercepted by a surface is more correctly calculated as:

Direct beam radiation x solar index + diffuse radiation x sky-view.

Total incoming radiation can be partitioned into its direct and diffuse components and sky-view factors calculated using function in the microclima R package (Maclean et al 2019 Methods Ecol Evol, 10: 280-90) which is also a better reference for the solarindex function than the citation given.

*SM: Dear Dr. Maclean, firstly we want to thank you for your appreciation of the manuscript and the useful suggestions. Following your advice, we i) explicitly refer to downward shortwave solar radiation throughout the text, and ii) calculated sw radiation as the sum of hourly $direct * solarindex + diffuse * skyview$.*

The EUMETSAT CM SAF radiation product (implemented in the microclima R package) has a limited geographical coverage (longitude 70°W to 70°E), and this hampers its use in key areas of the world (e.g. most of the Andes, of W North America and of Himalaya). Therefore we implemented the ERA5-Land radiation product (available at https://developers.google.com/earth-engine/datasets/catalog/ECMWF_ERA5_LAND_MONTHLY_BY_HOUR - nominal resolution: 11,132m).

*The ERA5-Land product has some advantages, given it provides monthly-averaged hourly estimates of both total downward ("surface_solar_radiation_downwards") and net radiation ("surface_net_solar_radiation" - i.e., directly accounts for albedo), and also provides albedo at the same temporal resolution ("forecast_albedo"). The ERA5-Land product does not provide the direct-diffuse separation (so, for both radiations and albedo, estimates refer to the sum of direct **and** diffuse portions). To overcome this issue, we implemented a direct-diffuse separation method. Following Gueymard & Ruiz-Arias 2016 (Solar Energy 128: 1-30), we choose the Yao2 model as it is the model showing the best performance in high-elevation and high-latitude (i.e., high-albedo) areas.*

Consequently we: i) used total radiation to estimate hourly clearness index K_t ; ii) used the Yao2 polynomial model to obtain the hourly diffuse solar radiation fraction K_d ; iii) estimated hourly diffuse and direct **net** radiation (MJ m^{-2}) by multiplying **ERA5-Land net** for K_d and $(1-K_d)$, respectively; iv) obtained effective hourly radiation implementing the same approach used in the `microclima::shortwavetopo()` function. Additionally, we followed the approach reported in Maclean et al., 2019 (*Methods Ecol Evol*, 10: 280-90) to (further) separate diffuse radiation in its “isor”, “cizr” and “refr” components (isotropically and anisotropically distributed and reflected back, respectively), considering the ERA5-Land albedo product cited above as the mean percent of radiation reflected within the study area (i.e., the albedo from adjacent surfaces - *albr*). Cumulative shortwave radiation was finally obtained as the sum of hourly estimates of the direct and diffuse fractions, following your suggestions (i.e., multiplying effective radiation by solarindex and skyviewtopo outputs). To conclude, we note here that calculating Clearness index (K_t) on the total radiation and applying it to the net radiation involves approximating the effect of albedo as constant between the two components (i.e., considering the separation as independent on the albedo). Furthermore, we changed the reference as suggested.

Doing it this way would omit the need to include cloud cover as a term in the statistical model, thereby simplifying it significantly.

SM: As suggested, we implemented the new approach and cloud cover is no more considered.

It would be more appropriate also to calculate absorbed rather than intercepted radiation by incorporating surface albedo ($\text{absorbed} = (1-\text{albedo}) \times \text{intercepted}$). There are global albedo datasets, but a simple framework would be to assume a surface albedo of ~ 0.15 without snow cover and ~ 0.9 with snow cover. This could further simplify the statistical model by omitting the need to include interactions between snow and radiation and cloud cover and snow.

SM: To take into account this issue, we used both albedo-free estimates of radiation and introduced background albedo using the `forecast_albedo` from the ERA5-Land product.

Note also that the main reasons that tree cover have an effect on ground temperatures is because it intercepts shortwave radiation, so it should really be included as an interaction term with radiation (as opposed to an additive term). A simple way to handle this pseudo-mechanistically would be to apply a correction to shortwave radiation prior to performing the terrain correction. Shortwave radiation below canopy is given approximately by $(1-\text{fractional canopy cover}) \times \text{shortwave radiation above canopy}$. There is a secondary effect of trees on surface roughness, which in interaction with wind speed controls the slope of the relationship between radiation temperature offsets and net radiation.

SM: Following your suggestion we introduced an interaction between shortwave radiation and tree cover. The direction of the effects of both tree cover and the interaction itself is correct, with areas shaded by trees showing lower temperatures, and a less steep response to shortwave radiation. However, the interaction term is not statistically significant, perhaps because the areas we are modelling are substantially treeless (only 1.7% of sensors are close to trees - percent coverage derived

from the Hansen's map is 44, 49 and 79%), and such limited number of sensors strongly reduces the statistical power for this interaction.

Overall, by going slightly further down the mechanistic route of modelling microclimate, the authors could actually significantly simplify their statistical model without adding much computational penalty, with the added advantage of giving their models greater physical meaning. The route to do this is to recognise, following Bennie et al (2008 Ecol Model 216: 47-59 and the Maclean ref given above that

$$T_{\text{micro}} = T_{\text{meso}} + d \cdot (R_{\text{net}} - L - G)$$

where T_{micro} and T_{meso} are microclimate and mesoclimate temperature respectively, R_{net} is net radiation (including the longwave component, which would need to be handled sensibly), L is latent heat and G , is the rate of heat storage in the ground and d a measure of the degree of thermal coupling between the surface and the air, a function of wind speed and surface roughness. By mesoclimate I mean macroclimate corrected for elevation and e.g. distance to glacier. G and L can be considered small and proportional to net radiation so the temperature offset from macroclimate (ignoring mesoclimate effects) is a linear function of net radiation, the slope of which is determined by d . Since d scales linearly with $\log(\text{wind speed}) \times \text{vegetation height}$, there is a direct route to calculating microclimate, though one would still need to account for some of the mesoclimate effects and snow cover. There are slight issues in applying this approach to monthly data, as is generally used when modelling using sub-daily time-steps, but since most of the relationships are linear, issues associated with Jensen's inequality only manifest in the calculation of d using $\log(\text{wind speed})$. However, I suspect this issue is quite minor and a nice model could be derived using this approach. Nevertheless, I leave it up to the authors to decide whether following this approach might simplify their model and make it more robust.

SM: We agree in principle that it would be great to move toward "the mechanistic route of modelling microclimate". As you state, working on the offsets (i.e., $(T_{\text{micro}} - T_{\text{meso}}) \sim d(R_{\text{net}} - L - G)$) would allow simplifying our model by several terms and interactions.

According to the Reviewer suggestion, the estimation of d requires information on vegetation cover, vegetation height and windspeed. Unfortunately, we are not aware of datasets providing fine-scale information on vegetation height and wind speed across multiple mountain areas of the world. We highlight that mountains are highly heterogeneous environments, where we can observe a huge variability of vegetation cover and height in just a few hundred meters (bare soils to alpine grasslands to forested areas). Similarly, windspeed has a great influence on heat, but in mountain areas it shows very strong variation even between nearby areas. Modelling the daily changes in speed and direction of wind (i.e., general patterns of movement of air masses + local effects of orography on these patterns + anabatic winds during the warmer hours + katabatic winds) at such small scale is extremely challenging, also considering the scarcity of direct information on wind speed.

As the Reviewer left it up to the authors to decide whether this approach should be followed, and considering the challenges of obtaining the relevant data, we preferred relying on the previous model (after integrating the improvements suggested by the reviewers).

Aside from this set of comments about the modelling, most of my comments are quite minor:

Justification for calculating mean as opposed to say monthly maximum and minimum temperature is not

really provided, and I wonder if by not doing this they are missing some interesting patterns. Certainly one might expect spatial heterogeneity in maximum temperature to be more pronounced than in mean temperature and the extremes may also be more biotically important.

SM: Following this suggestion, we assessed the pattern of minimum and maximum temperature (calculated as the 5% and 95% quantiles of all monthly values following Lembrechts et al., 2022 - Glob Change Biol. 28:3110–3144.). Considering the training and validation datasets, minimum and maximum temperatures showed very strong correlation with the mean monthly temperature (Pearson’s correlations: $r > 0.91$). This suggests that adding the analysis of min-max temperature would not add significant information. We expanded the Methods section (lines 704–710) to clarify this issue.

Lines 48-50. Potentially slightly misleading as this presumably is largely to do range overlap meaning that any 10% of the planet would host >10% of species (rather than it being the case that mountains are particularly diverse).

SM: This sentence highlights the fact that many mountain areas are biodiversity hotspots, i.e. they host a much higher biodiversity than expected on the basis of their area. For instance, several of the classical “biodiversity hotspots” match mountain areas of the world (e.g. Myers et al. 2000 Biodiversity hotspots for conservation priorities. Nature 403, 853-858), and “with about 25% of all land area, mountain regions are home to more than 85% of the world’s species of amphibians, birds, and mammals, many entirely restricted to mountains.” (Rahbek et al., 2019, Science 365: 1108-1113). To clarify this point, we modified this sentence of the Introduction, explicitly focusing on the estimates by Rahbek et al., 2019.

Lines 63-64. Consider replacing ‘topography’ with ‘terrain’

SM: Replaced.

Lines 67-68. More pronounced just above rather than below. But that doesn’t negate the point you are making.

SM: Thanks! We modified to “The decoupling between micro- and macroclimate is particularly pronounced near and, to a lesser extent, below soil surface^{25,26} ...”.

Lines 68-70. Worth mentioning elevation lapse rates here?

SM: Added.

Lines 81-83. Not quite clear whether you are talking about spatial or temporal buffering here, but actually

temporally near-ground temperature has greater variance than the shade temperatures measured by weather stations for reasons the authors themselves outline in lines 68-70. Not true under snow cover though.

SM: The paragraph started with “At fine spatial scales”, and it was thought about spatial buffering. Following your point, we modified the sentence to “The spatial buffering of microclimate is the dampening of macro-climatic fluctuations due to local conditions (e.g. topography and vegetation cover), such that these fluctuations still exist at the microclimatic scale, but have lower intensity and a more limited effect on organisms³⁴.”.

Lines 489-506. It would be useful to know which devices were used to obtain temperature measurements as different devices have different degrees of accuracy (though wildly inaccurate results is much more of an issue when measuring air temperature in sunshine). It would also be helpful to be reassured that ‘devices’ were located in manner that captures the range and appropriately stratifies determinants of microclimate (though I recognise that there are significant logistical challenges here).

SM: To clarify this issue, for each device / station, a series of summary information are provided in Supplementary Table 1 [including device brand and model, period of recording (start and stop), number of recordings and recording frequency, as well as depth of burying]. All devices were buried far from sources of possible shadows at the micro scale (e.g., far from boulders), but depending on the scope of the different projects during which data were collected, devices were buried at different depths. We modified the sentence to highlight where the key information is available.

I hope my comments are useful

Ilya Maclean (I have a policy of signing my reviews where they make reference to my own work)

SM: For sure they are...they've been fundamental. The integration of all these precious suggestions allowed key improvements to our model! Thanks again!

RESPONSE TO COMMENTS OF REVIEWER #2

This research uses an extensive network of temperature sensors which have measured soil temperatures in the areas surrounding glaciers to attempt to quantify the large microclimate variability in such regions and evaluate the consequences for the foreland ecosystems in the context of climate change. The complex topography, coupled with high elevation and variable snow cover, means exposure to (often intense) solar radiation is extremely variable, which can create strong instantaneous temperature gradients and high amounts of spatial variability. A model is developed to predict monthly soil temperatures based on such local factors and then applied globally to model changes in microclimate variability over the last 20 years. The main findings of the paper are that a) changes in microclimate have been greater nearer glaciers than further away and b) that local temperature variability in the vicinity of glaciers is extremely large and outweighs possible climate change forcing, at least at the current time, meaning that ecosystems may have

a buffer to withstand some of the more detrimental effects of rapid global warming.

In one sense, the paper tells us something we already know, since the idea of refugia (created by strong microclimates) is not new. However, the dataset I think is unique and the modelling methods are quite comprehensive and the model fairly successful, at least in statistical terms. I am not an expert in all of the statistical methods, but the models appear robust and methods are taken to assess their reliability. The paper is well written and referenced and there is a clear understanding of the relevant literature which sets the work in an appropriate context.

SM: First of all, we would like to thank you for your appreciation of the manuscript and the really useful suggestions: facing your criticisms allowed us to strengthen the results and, in our opinion, greatly improved the manuscript.

There are some limitations. Despite being applied globally, the model is not really global because of the limited and uneven concentration of data which goes into it. Although field data from 26 glacier forelands are included, 21 of the sites are from the Alps, and only 3 from Peru (tropical) and 2 from Svalbard (arctic). Thus, the model is dominated by mid-latitude maritime conditions. Taking out one glacier at a time to validate the model still means all runs are still Alpine dominated and I would therefore not expect the model coefficients to vary much (when using 26 rather than 27 locations) so the conclusion that the model is robust based on this alone is making a bit of a leap of faith.

There is little analysis to convince readers that it works just as well in the “minority” locations as it may do in Alpine locations. Obviously, the best solution is more comprehensive data, but I realise that it probably is not available. At the very least therefore it would be important to look at specifically how the non-Alpine forelands are predicted by the Alpine dominated model? i.e. does it work in the tropics or in the Arctic just as well as on the other glaciers?

Beyond the “leave one out” approach, it might be helpful to develop models separately for the three latitudes to see if they are different or similar (I realise sample sizes are uneven but that is unavoidable). Similar coefficients for the tropical glaciers for example will increase confidence that the model is indeed globally applicable. Different coefficients will cast light on variable microclimate controls. The mid-latitude Alps are also fairly maritime, and continental/arid glaciers such as those on parts of the Tibetan plateau, or at extremely high elevations in the tropics (e.g. Kilimanjaro) are missing from the model. It would be great if it could be validated on such a contrasting example. So in summary, at the moment the paper requires some more critical analysis to show that when the model is extrapolated globally into regions with quite different climatic regimes, that it is reliable.

SM: Following your advice, we looked again for sources of independent validation and finally found the SoilTemp dataset, which is suitable as an external validation dataset and has a very broad spatial and environmental coverage.

We started from the total (and huge - congratulations to Dr. Lembrechts and the SoilTemp team!)

SoilTemp dataset (available at <https://doi.org/10.5281/zenodo.4558663> -

20201215_SoilTemp_monthly_cleaned.csv), which includes 380,676 monthly recordings.

Within SoilTemp, we selected the sites with i) Height (i.e., depth of burying) within the range -15 to -5 cm and ii) distance from glaciers (measured using the GLIMS glacier outline) $\leq 3,000$ m. This reduced the dataset to 7,052 monthly recordings. The validation dataset was further cleansed by removing duplicated records (same coordinates, Height, month and year), and all stations on water bodies / rivers or human infrastructures (i.e., roads). This further cleansing reduced the dataset to 6,505

recordings from 170 stations, distributed in Europe (Alps, Pyrenees and Scandinavia) and the Himalayas (India and Nepal). For each record, we extracted environmental variables **for the year and month to which observations refer**, using the coarse-scale global approach (i.e., EarthEngine data at 90 m resolution), and generated predictions using the mean coefficients estimated under the leave-one-out approach. Given that here we use estimates of snow-free days (sfd) based on NDSI-data, we excluded all records for which the estimate of percent sfd was based upon less than 15 images (to reduce noise). Additionally, as in the previous version, we excluded all records for which the estimated sfd was shorter or equal to 20%. We highlight that there was no calibration point in Asia, thus this is a fully independent validation dataset. We then compared observed and estimated monthly averages using, as in the previous version, weighted R^2 weighted Mean Absolute Error and weighted Root-Mean-Square Error. In the independent dataset, weights were calculated as to assign to each region (Europe and Himalaya) a weight of 1, and within each region to give equal weights to each sampling station and snow-free period (a global proxy for the month, in our view). We identified sampling stations based on pairwise geographic distances, clustering in the same station all devices closer than 1,000 m each other. We used this approach because the number of records was highly variable across areas, and we wanted to avoid an excessive weight of a few sites. The external validation dataset confirmed the excellent predictive performance of our model ($wR^2 = 0.855$; $wMAE = 2.214$; $wRMSE = 3.000$). The new Extended Data Figure 2 provides an overview of the model performances for both the training / internal dataset and the independent one.

To further take into account the Reviewer comment, and apart from the leave-one-out approach, we reviewed the approach used to calculate weights in the internal validation. Similarly to what we explained above about the external dataset, we assigned to each region (Polar, Mid-latitudes and Equatorial) a weight of 1, and assigned equal weights to each glacier and snow-free period within each region. With this approach, the goodness-of-fit statistics now effectively account for the unbalanced sampling distribution.

To conclude, even if the additional independent dataset does not have (again) a true global coverage, in our opinion it allows evaluating the model behaviour in places with completely different climate regimes, and seems to support the use of model coefficients to map projection globally.

There is also some limitation temporally. The data set has 2203 monthly readings from 175 sites which means an average of 12.6 months from each site. I realise these are not consecutive because they represent snow free periods, but even so this is not many years at each site, and they are for different years at each location. Assuming a typical snow free season is 4-6 months then we have about 2-3 summer seasons on average at each site. So variability between seasons and years (which could also be different between each location) must contribute to the results and success of the model. I understand all this is difficult to quantify, but it is important to discuss.

SM: We agree that interannual variability does exist and might affect our results. In fact, significant components allowed us to account for the inter-annual variability. Our revised model takes the form:

$$\text{soilT} \sim mT + rad + clo + sfd + dg + \text{additive effects}$$

$$sfd:mT + sfd:rad + sfd:dg + \text{sfd - interactive effects}$$

$tc:rad +$	tc - interactive effect
$d:sfd +$	d - interactive effect
$tc + pf + d +$	intercept corrections
$1 gl$	random intercept

In this model, three variables (mT, rad and sfd) have been obtained from global time-series to have direct measures of temperature, solar radiation and snow-free days at monthly scale (i.e. each month within each year has a different value of mT, rad and sfd) and, for sites sampled for more than one year, we included separately the measurements of a given month taken in different years. This allowed us to successfully take into account the variability between years and seasons. We also highlight that, in our model, mT, rad and sfd were the variables with the largest contribution (see the new Extended Data Figure 1).

Overall, even if our data does not allow considering the full potential variability at the local scale (we could obtain this information only with long-term monitoring programs), our model creates direct links (the model coefficients) between soil temperature and the variables representing the inter-annual variability (macroclimate, season length, shortwave radiation, and their interactions). Following the Reviewer suggestions, we expanded the Introduction and Discussion sections to highlight the importance of inter-annual variability (lines 98-102 and 354-355).

I also have some minor comments/concerns (see line numbers)

Line 67: Microclimate variance can be extreme above and at the surface, but actually just below a lot of this variability is reduced since the soil temperature shows a damped response. So it is a bit misleading to group near and below surface together (they are very different).

SM: Following this suggestion, and also following comments of Reviewer 1, the sentence was modified to "The decoupling between micro- and macroclimate is particularly pronounced near and, to a lesser extent, below soil surface^{25,26} ...".

Line 128: Cloud cover has a greater effect when there are fewer snow free days (i.e. more snow). This does not make physical sense (surely more snow means a weakened response to radiation). It could be because solar radiation is typically strongest in late spring/early summer when snow is still melting (assuming northern hemisphere here)? So the relationship at the monthly scale is due to this seasonal association and not any physics? Perhaps this just needs a bit more clarification.

SM: Following the suggestion of Reviewer 1 cloud cover was now removed.

Figure 2: The axes have strange irregular units.... i.e. -0.69, 2.65, 6 etc for soil temperature... they need to have regular intervals. Some axes do not have units at all (i.e. solar radiation). Also why are individual values given for high, low and medium snow free days and cloud cover (surely they should be ranges?)

SM: In the previous version of Figure e.g., 2a (that to which you refer) the units are not irregular, the apparent irregular pattern was related to automatic rounding of axes performed by R. The issue was fixed in the new version of the manuscript.

We also fixed the issue of lacking units for solar radiation.

Regarding “individual values”: this is the typical way to plot interactions used, e.g., in the visreg R package. In order to plot interactions, the most frequent approach is to choose 2-3 values to condition the relationship between two continuous variables (the response and one predictor) on the effect of a second predictor. Imagine you have $y \sim \beta_0 + \beta_1x_1 + \beta_2x_2 + \beta_3x_1x_2$, being y , x_1 and x_2 continuous. To plot the fitted $y' \sim x_1$ you will have to calculate $\beta_0 + \beta_1x_1$, but you are neglecting the effect of $\beta_3x_1x_2$; to include also this effect remaining in 2D, you will have to calculate $\beta_0 + \beta_1x_1 + \beta_3x_1x_2$, at chosen values of x_2 (imagine 25, 50, 75 - i.e., $\beta_0 + \beta_1x_1 + \beta_3x_1 * 25$).

This approach enables us to show the relationship $y' \sim x_1$ at low, medium and high values of x_2 . We expanded the caption of Figure 2 to clarify how interactions are visualized.

L 145: Downscaled macroclimate: it is not clear what this is. I think this is just TerraClimate altered with a standard lapse rate? I suppose the TerraClimate grid cell elevation may be very different from the foreland/datalogger elevation and so that is why a correction is required? If so I think this needs to be explained. The other question is how reliable this lapse rate correction is. As we all know, lapse rates are very variable in space and time, and I would think this might be a major source of uncertainty – but there is no sensitivity analysis of the decision to use a fixed lapse rate.

SM: Yes, the downscaled macroclimate corresponds to TerraClimate time-series altered with a standard lapse rate. We expanded the text to clarify this point.

We fully agree that lapse rate shows spatial and temporal variability. Following Maclean et al. 2019 (Methods Ecol Evol. 10:280-290), apart from a series of known constants, lapse rate calculation requires knowledge of the reference temperature (eventually known from TerraClimate), but also of **specific humidity and atmospheric pressure**.

Unfortunately, detailed information on the spatial and temporal variation of lapse rate and its drivers at the global scale is lacking. ERA5-Land provides estimates of surface pressure and dew-point temperature at coarse resolution (11 km resolution), but the authors specify “The strong variation of pressure with altitude makes it difficult to see the low and high pressure systems over mountainous areas...”. Thus the available global-scale data (ERA-5) are not appropriate for our study. As a consequence, we believe that attempting to estimate lapse rates without robust and high-resolution data on specific humidity and atmospheric pressure would imply introducing noise / errors possibly larger than those produced by a fixed, but standard lapse rate value.

To understand how the approach is sensitive to the chosen lapse rate, we rerun the model using alternative values. Rolland 2003 (J. Clim 16.7: 1032-1046) reports several values for European mountain environments (mean temperature - annual: 3.9 to 8 °C / km), and estimates values in the set [5.4 ,5.8]. Similarly, Minder et al., 2010 (J Geophys. Res. Atmos, 115.D14) report values ranging between 3.5 and 6.1 °C / km for the Cascade Mountains of North America. Based on these values, we decided to recalculate downscaled macroclimate using fixed values of both 5 and 8 °C / km, rerun the model and obtain summary statistics. Encouragingly, the results obtained with different lapse rates were strongly consistent:

	5 °C/km	6.5 °C/km	8 °C/km
--	---------	-----------	---------

wR2	0.8386879	0.8473596	0.851961
wMAE	1.51971	1.470633	1.441974
wRMSE	1.887678	1.83563	1.807438

To take into account the Reviewer suggestions, we thus expanded the explanation of the downscaling procedure in the Methods section, with a clear reference to the standard and fixed lapse rate value used and we now acknowledge the issue of its spatial and temporal variability (lines 543-549: “To produce high-resolution macroclimatic surfaces while accounting for the adiabatic decrease of temperature, we downscaled monthly temperatures applying a fixed environmental lapse rate of $-0.0065^{\circ}\text{C}/\text{m}^{55}$. To understand how sensitive the downscaled surface to the chosen lapse rate is, we also produced estimates using different lapse rates (-0.005 and $-0.008^{\circ}\text{C}/\text{m}^{56}$). Estimates obtained using the different lapse rates showed very high pairwise correlation (minimum Pearson’s $r > 0.98$), thus we decided to use the standard value of $-0.0065^{\circ}\text{C}/\text{m}$ as suggested by Barry⁵⁵.”). We only reported the correlation between the predictors within the manuscript (i.e., we avoided comparing the model outputs obtained with different lapse rates) in order to keep the text concise and easier to follow. Nevertheless, if the Reviewer believes it is important, we are available to integrate this additional table as supplementary information.

Line 250: The lack of change in snow cover season length as you move away from tropical glaciers is interesting and yes does imply that it might be due to lack of snow cover (so lack of snow cover change). However, in the tropics there is limited seasonality (at least due to temperature) and in many cases snow falls intermittently throughout the year (mostly in wet seasons) only then to disappear in days/weeks so there is no well-defined snow season in the first place. I wonder whether it is not clear what you are measuring.

SM: For Fig. 3c, the duration of the snow-free season was calculated as the number of days with no snow on the ground for the whole year, averaged for each of the two 5-years periods. Consequently, this measure is expected to be appropriate also for tropical areas, where there is limited seasonality of temperature and low persistence of snow on the ground. To clarify the point, we expanded the explanation of how the duration of the snow-free season was calculated. Specifically, we added a sentence at lines 772-775 “For the same set of points, we also extracted the annual duration of the snow-free season for the two periods. We measured the snow-free season as the total number of days with no snow on the ground (i.e. with fractional snowcover $< 40\%^{62}$) during the whole year, averaged over each of the two periods.”.

Line 286: Change to "temperature increase is often faster"..... (not always I would argue).... On average positive EDW has been shown to exist, but there are places at high elevation which are not warming faster.

Line 311: Change to “Although we tried....”

Line 316: Change to “in others they are retreating considerably”

Line 319: Change to “Our analysis provides....”

SM: We performed all the requested modifications.

Extended Data Tables 1 and 2: This should be largely understandable from the caption alone and so a bit more information about how the coefficients were determined would be useful. In Table 1, what are d(10) and d(15) – dummy variables?

SM: Following this suggestion we added several more info.

Line 527: It is strange that cloud cover is taken from a different time period (2000-2014). Why? Would it not be better to use the same period as the soil temp data?

SM: In the current version cloud cover was removed following comments of Reviewer 1. Still, a simple justification is needed: in the previous version we used a published dataset that unfortunately has a different temporal extent. We believe that the new model (not including cloud cover) is more appropriate.

Line 530 ff: Snow cover – I know you can get this from the logger temperature amplitude but would it not be better to use MODIS NDSI anyway (since this is transferrable) in the original model. At the moment I think you use it to determine the threshold amplitude, but then define snow days using the temperature and not NDSI. This might be more accurate for this model, but eventually in the global model you have to use NDSI anyway (since real temperatures are unavailable) so this just creates differences between the model which is tested and calibrated and the one used for global extrapolation.

SM: The Reviewer is right. During the model calibration / training we used temperature-based estimates of season length, while we switched to NDSI-based when projecting at the global scale. We had lots of thoughts on this point, given that directly implementing NDSI-based estimates increases model transferability, and simplifies the overall approach. On the other side, the main aim of the analyses was to disentangle the drivers of soil temperature in proglacial areas and to obtain a robust model. Additionally, as explained before about the external validation, single-month estimates of NDSI severely suffer from the lack of complete series. So, we would have to either use multi-year averages or exclude from the calibration dataset samples with inadequate image coverage, and in both cases the estimates would be less accurate. We now expanded the text at lines 590-607 to clarify the issue.

Line 558/559 and 661: Tree cover is not really binary is it? Particularly in the glacier foreland where isolated trees might be common. The text implies it was yes or no, but surely a percentage value would have been more appropriate. There is lack of detail on this. What was the threshold?

Permafrost: Line 561 ff: What depth are we talking about when we define permafrost? I suspect this is at a much deeper location than the temperature loggers, and if so, I am not sure what the relevance is, again of a binary variable.

*SM: About **tree cover** you're right, it is a percentage in the original dataset, but we converted it to tree occurrence for all the cells with more than zero coverage. We converted it to binary as the number of stations with tree cover > 0 was very small (just three), thus we did not have enough variability of tree cover to model linear relationships (we had too many zeros).*

About **permafrost**: the author of the dataset (Gruber 2012; *The Cryosphere*, 6, 221–233) did not specify the depth at which he worked, but defined permafrost as “sub-surface material (excluding glaciers) having a temperature of less or equal to 0 °C during at least two consecutive years (ACGR, 1988)”. We think that in this publication the author was simply interested in the spatial dimension of the phenomenon (i.e., mapping permafrost extent).

Line 669: Change to “instead of”

SM: Modified.

Line 671: How many points in each cell (on average?). I don’t understand later how there can be fewer than 9 points in a 50 by 50 km cell?

SM: For each 50km cell, the procedure was: generating 10 random samples at given distances from glaciers (2 for each distance class), then checking if they all have the relevant information. Owing to the random nature of the selection of sampling sites, sometimes it happens that points at a specific distance from the glacier fall e.g. in the sea, and / or the associated temperature/ radiation / NDSI data are NA, resulting in less than 10 points.

In the revised version of the analyses, we only retained cells with complete information (10 points), in order to obtain comparable and robust estimates of global changes, in our opinion. We modified the text to clarify this issue (lines 767-769).

To summarise I think there are some valuable insights here, but the main weakness is the uneven distribution of data which goes into the model (basically the Alps dominate the model), and the assumption that it can be clearly extrapolated globally with limited error. More analyses to support this contention and a broader discussion of applicability to different climate regimes is required. Other than this I think the paper provides potentially valuable insights and I would encourage the authors to make appropriate revisions.

SM: Thank you again for your thoughtful insights and comments on the manuscript. We think that the changes we made following your suggestions greatly improved the manuscript.

RESPONSE TO COMMENTS OF REVIEWER #3

In this paper, the authors model the soil temperature in glacier forelands using data from three mountain climatically-distinct mountain regions across the globe. They disentangle the drivers of these soil temperatures and use their model to assess soil temperature changes globally over the last two decades. While I applaud this project – quantifying microclimate change is urgently needed, especially in vulnerable environments like mountains – I have a few comments that need clarification.

SM: Dear Dr. Lembrechts, first of all we want to thank you for your appreciation of the manuscript, and the fully embraceable claim to the particular urgency of quantifying microclimate changes in the rapidly changing mountain environments.

Most importantly, the global coverage of points for model calibration is limited to three mountain regions, and I think that's a bit limited. The authors acknowledge this in passing, but don't take further action in this regard. What comes to mind as things that I find missing most are: 1) out-of-sample model validation: would your model be able to predict temperatures accurately in another mountain region? And 2) a masking of areas outside the model parameter space (with this I mean: points that have environmental (and especially climatic) conditions not within the range of any of your data points should be masked from the global assessment, as here it becomes increasingly hard to trust model extrapolations. For the first point, I don't think the leave-one-out method is sufficient, as you always keep all three your mountain regions in the model. You might really have to look for another dataset from another mountain region (or test the predictive power of the model if you for example remove the whole of Svalbard. This would obviously be drastic, but the same could be set of using these three mountain regions to predict what happens in the Himalayas without verification. For the second point, at least checking the environmental space of your observations and the rest of the globe, so a reader can assess if this extrapolation might be problematic, would be a good start.

SM: Following this suggestion, and also considering the suggestions of Reviewer 2, we implemented an external validation dataset based on "your" SoilTemp dataset (once again, congratulations to you and the SoilTemp team for the huge effort made!!). To effectively test model transferability (also considering the comments of Reviewer 2), we decided to keep our dataset separated from the SoilTemp one. Specifically, we used our data for model calibration (using the leave-one-out approach to estimate average model coefficients), and the SoilTemp dataset as an external, independent validation dataset. We additionally changed the way weights are calculated for the internal R^2 , MAE and RMSE, so that now the weights from each region (i.e., Polar, Mid-latitudes and Equatorial) sum up to the unity - i.e., they equally contribute to goodness-of-fit statistics. This allows limiting the excessive weight of records in specific areas (e.g. the European Alps). Our model showed very good performance with the validation dataset ($wR^2 = 0.855$, $wMAE = 2.214$ and $wRMSE = 3.000$; see Extended Data Figure 2), strongly suggesting that our model has good transferability.

Extrapolation issue

In the revised version of our manuscript, the model was developed on a training dataset (our 175 sensors), validated on 161 SoilTemp sensors (validation dataset) and then projected globally. As the validation showed good predictive performance, we assume that major problems would arise when the model is projected to areas with conditions outside the ones experienced during the training or validation.

Therefore, in order to assess extrapolation, we tested whether independent variables used for global projections have values falling outside the range observed for training and validation, an approach often used in analysis of global change biology (e.g. Elith et al. 2010 Methods Ecol Evol 1: 330-342). The majority of independent variables (i.e., distance from glacier, permafrost and tree occurrence, percent of snow-free days) did not show extrapolation issues, as their values were within the range experienced during training.

Some extrapolation occurred for temperature and shortwave radiation, still the extrapolation levels were very limited. For temperature, only the 0.67 and 0.75% of values were outside the range (for 2001-2005 and 2016-2020, respectively). For shortwave solar radiation the percentages were slightly higher, but still small (2.35% and 2.34%, respectively). This is not surprising to us, as our predictions were limited to particular areas (glacier forelands) characterized by very specific conditions. We expanded the Methods section (lines 787-796) to provide a quantification of the extrapolation issue.

Specific comments:

L37: make clear that this 1-to-10 times is about the warming that so far has happened, and not about 'climate warming' in general – a reader who only reads the abstract will otherwise miss this nuance, I think. The climate warming that happened so far is significantly smaller than the climate warming to expect in the near future, even in the best case scenarios. You mention this in the discussion, but in the abstract the nuance is easy to overlook.

SM: We fully agree with this comment and modified the manuscript accordingly.

L64: perhaps say 'partially decoupled'?

L84: given that you just defined 'buffering' in the microclimate context above, I would use another term here. 'Limit'?

SM: We included the suggested changes.

L84: not sure this concept only works for moving individuals, I think this is often (more often?) considered for populations

SM: For sure, but in the strict sense movement is made by individuals, and this allows populations to colonize new areas. We replaced "individuals" with "organisms", so that we can include both individuals and the subsequent impacts on populations.

L90-104: I would here already explain how glacier retreat/advance is incorporated in your temporal analysis, as the fact that you focus on glacier foreland microclimate change makes that a key parameter and its not explained here that this is incorporated.

SM: Added.

L103-104: how did you make this comparison with other high-elevation environments? Perhaps specify the used methodology here.

SM: This referred to the comparison of the effect of change at different distances from glaciers (i.e., mainly to Fig. 3 and Extended Data Figure 6). We rephrased to make the concept clearer.

L120: is this also already downscaled macroclimate? The first mention of downscaling comes much lower. Perhaps also say that this is lapse rate-based downscaling and not including any other meso- or microclimate parameter.

SM: Yes, it is. We rephrased to account for both of your suggestions (lines 124-125, "Soil temperature was positively related to macroclimatic temperature (downscaled using lapse rate, see Methods), downward shortwave ...").

Fig. 2g: one would expect a significant interaction between depth of burial and seasonality, with deeper sensors showing higher temperatures in autumn and towards winter, and shallower sensors showing higher temperatures in spring and summer, no? Yet this was not tested, I understand.

SM: Following this suggestion, in the revised model we included the interaction between seasonality and depth of burial (Fig 2c). As expected, deep sensors show higher temperature at the beginning / end of the season (small percentage of snow-free days - sfd) but lower during summer (i.e., temperatures are more "stable" during the year). On the contrary, shallower sensors are more sensitive to changes in air temperature, being colder with small sfd and warmer during summer.

Fig. 2: the strong interactive effects of snow free days makes me ponder. As you used some kind of truncation approach of snow free days, how much of that is still just the buffering effect of snow?

SM: Snow certainly has a buffering effect, and we expect a larger buffering with less snow-free days. In these environments, removing months with partial snow cover would be problematic, as partial snow cover occurs in 34% of records. This interaction allows taking into account the dynamics occurring during these months.

L164: is the MAE in °C?

SM: Yes, it is. Specified.

L175: I'm not entirely sure it's meaningful that your model – calibrated on your in-situ measurements – is more strongly correlated with those in-situ measurements than any other product not related to your in-situ measurements. This validation feels rather circular to me. This would make more sense to me, personally, if done out-of-sample on an independent dataset.

SM: Following this suggestion and following suggestions by Reviewer 2, in the current version of the manuscript we implemented an external validation dataset. We also simplified the comparisons excluding the downscaled macroclimate and produced a new Extended Data Figure 2 with the same comparisons between observed and predicted - TerraClimate - Chelsa for both the training and testing datasets; our model remains the best-performing one.

Extended data figure 2: here, to me, having the Root Mean Square Error – which gives higher value to extreme outliers – would be more logical to show the difference, no?

SM: We introduced wRMSE as further goodness-of-fit statistic in both the EDF2 and throughout the text.

Fig. 3: what to me seems more important than absolute changes in temperatures is how much these temperature changes differ from macroclimatic observations of climate change. Trends will be the same, but it allows a reader to put things in a better perspective.

Fig. 3a: would it make sense to zoom in on one or a few mountain region(s) as an example?

SM: Following both suggestions, we now introduced two Extended Data Figures, aimed at showing some spatial patterns at higher resolution (EDF3), and the difference between soil and macroclimatic changes (EDF4). To this aim we first calculated offsets between coarse-grained TerraClimate and model predictions during the snow-free season, following Lembrechts et al., 2022 (Glob Change Biol. 2022;28:3110-3144). We then calculated the difference between offsets, obtaining the change in soil temperature once removed the effect of macroclimate change.

Regarding the distribution of offsets, it is interesting to note that for both the 2001-2005 and 2016-2020 periods, they're mainly positive with a mean of 3.5 and the 25 and 75 quantiles around 0.9 and 6.2, respectively. These values are very close to those reported by Lembrechts et al., 2022 for the mean annual temperature offset of the tundra biome (Fig. 1b T - mean \approx 4.5; 25 and 75 quantiles \approx 2.5 and 6, respectively).

We therefore expanded both the Results and the Discussion (lines 207-210 and 304-306) to highlight that detected temperature changes were not identical to macroclimatic observations of climate change. The difference between microclimate and macroclimate was particularly large nearby glaciers and in tropical areas, highlighting the particularly fast warming of these areas. In our opinion, this might be due to the increased length of the snow-free season, particularly pronounced in these zones and distance classes (Figure 3d).

Fig. 3a: it's surprising to me that Svalbard is excluded from your predictions, even though it's one of the three mountain regions where you have data, would it make sense to model for the whole globe and then use a mask to exclude areas out of the range of calibration?

SM: Unfortunately, this is due to lack of digital elevation data for high latitudes on EarthEngine. SRTM is limited to 60° N and S, and we had to integrate GMTED and SRTM to cover the 60 to 72°N area. Downloading and re-loading in EarthEngine all the ASTER dataset (those used in the training - 1 degrees tiles) for the whole study area would have been extremely time consuming.

Fig. 3c: somehow lacks resolution. Would some kind of density plots not work better?

SM: The figure was re-drawn with the new model, and the new version of the figure shows a more balanced distribution of T_{bp250} (due to the new approach to the calculation of shortwave radiation, cf. the response to Reviewer 1), and we hope the problem is solved.

Fig. 3d: based on the main text (L246-248) this seems satellite-based, is that correct?

SM: Yes, it is; we expanded the legend to clarify this point. Also following comments of Reviewer 2, we now better explained in the Methods how this duration was calculated. We added a sentence at lines 772-775 "For the same set of points, we also extracted the annual duration of the snow-free season for the two periods. We measured the snow-free season as the total number of days with no snow on the ground (i.e. with fractional snowcover < 40%⁶²) during the whole year, averaged over each of the two periods."

L194-204: the high amount of numbers in this paragraph make it difficult to process them

SM: The paragraph was simplified following your suggestions.

Extended Data Fig. 3: why is the spatial extent of the maps different in each season? I'm assuming the answer is on line 655, but that would best be specified here.

SM: The Reviewer is correct. There is strong seasonality in the occurrence of points with $sfd > 20\%$; if at least one point within a cell showed $sfd < 20\%$ for three months (as occurs e.g. in winter), the whole cell is NA, and there is no prediction. We explained the point in the legend.

L271: in your analysis, these aren't really trends, but spatial patterns, no? The temporal trends are just a space-for-time extrapolation that is not tested

SM: Our analyses are a combination of spatial (comparison of sensors on different areas) and temporal patterns (as we developed temporally explicit microclimate information), and both processes are relevant to our pattern, thus we modified the sentence to improve clarity. We highlight that, when projecting the model, we used the monthly temperatures recorded between 2001 and 2005 and between 2016 and 2020, averaged over each period. Consequently, even if we use two discrete steps, our model is based on the underlying recorded temporal change of macroclimate. To clarify the point, in the current version we repeatedly stressed the use of time-series throughout the text.

L302: the discussion of soil moisture comes a bit out of the blue, as this was no parameter in your analyses.

SM: Yes, we know, but that is another important parameter that we have to account for when analysing the effect of climate changes on soil biota. Following this suggestion, we shortened the paragraph and better integrated it into the text.

L309: abrupt switch to lack of global coverage from the soil moisture story. New paragraph, perhaps?

SM: Modified.

L329: I feel like you should explain the concept of the 'digital twin', why it is important. Ecosystems to me have so many more aspects than microclimate, so this might be a big leap to a digital twin of the ecosystem, here. It could be a 'digital twin' of the abiotic conditions or something, perhaps? Or I'm misunderstanding the concept of the digital twin...

SM: A "digital twin" is the virtual replica of a real system. The concept was born in engineering applications, when a virtual replica of a power, chemical, electronic, or mechanical industrial plant was needed to verify the effects of certain actions (e.g., a valve that breaks or a transistor that burns). Such concept was recently extended to the natural world, with the goal of building "digital twins" of the ocean or of the entire climate system, see e.g. Bauer et al., 2021 (Nat. Clim. Change, 11, 80-83) or https://www.esa.int/Applications/Observing_the_Earth/Destination_Earth. In this framework, a digital twin is characterized by a continuous flow of data from various sources (ground data from stations, satellite observations, drones, etc), and a blended ensemble of data-driven (usually based on Artificial Intelligence methods and machine learning) and process-based models. Such a goal is, in our opinion, quite visionary and perhaps a bit far-fetched for the climate system. However, it should be possible to build a working digital twin for a specific (eco)system, such as a pond, or a fragment of forest, but also for a type of environment. This is what we meant with the sentence in the paper. We slightly expanded the sentence (while keeping it concise) to clarify this point.

L489: how is a glacier foreland defined here?

SM: We focused on the definition of glacier foreland as "landscapes emerging after glacier retreat" (Zimmer et al., 2022 WIREs Clim Change 13:e753) and clarify this both in the Introduction and the Methods (lines 53-56 and 521).

L526: cloud cover has a coarse spatial resolution – especially for mountains, is it worth discussing this in the discussion as a potential reason for flaws?

SM: Following the suggestions of Reviewer 1, cloud cover is no longer considered as a predictor of soil temperature.

L566: was month not included as a factor in your models? Shouldn't it be, at least as a random effect to correct for unaccounted seasonal effects? Otherwise, it could be pseudoreplication as well. The latter is reflected in Fig. 1, where you are actually depicting sensors x months rather than sensors.

*SM: As explained before, we avoided using calendar month given the large spatial scale of analysis, as “month” makes little sense when comparing disparate areas of the world. When we include a random effect into a model, we presume that observations sharing the same value of the random effect are more similar than expected by chance. However, “June” in Svalbard is completely different from “June” in the Andes and so on, thus the use of month as a random factor would not have been appropriate. Instead, our model included *sfd*, that we considered as a satisfying global proxy for the same type of information.*

Problems of pseudo-replication would occur if, for the same station and month, we would have only one value of the independent variable in different years. This issue potentially occurred for cloud cover in the previous version of the manuscript. However, cloud cover was removed from the revised model, and was replaced by solar radiation, for which we used a different value for each month and year.

In the revised version of the manuscript, for a given station and month, all the variables showing seasonality have a different value for each month and year (i.e., are retrieved from time-series), and this should remove any issue of pseudo-replication (inter-annual variability makes data theoretically independent).

We modified the text (lines 539-542) to clarify this point (“TerraClimate was chosen as it provides time-series of monthly temperatures from 1958 to present, with yearly updates. This allowed relating soil temperature to macroclimate conditions in any given year and month between 2011 and 2021.”).

L581-586: isn't the effect you are trying to remove here not also exactly the one driving the interaction with snow free days in your model?

SM: This is partly true: the cleansing phase you refer to allowed us to i) obtain an acceptable distribution of model residuals and ii) remove the data for which relating climate or radiation to soil temperature is completely useless (all we know that temperature is almost constant due to the insulating properties of the snow cover, and substantially it does not depend on external inputs). After excluding these data, the interactions allowed quantifying the remaining effects (i.e., those of a partial snow cover).

L673: why 3100 meter? I thought you limited the analysis to 3000 m?

SM: Working with centroids of 90 m-resolution cells, those numbers approximate the classes of distance we are interested in: depending on where the cell centroid falls, that 200 m interval (2900-3100) allows sampling maximum two centroids in the requested range, but sometimes only one. It was just a workaround to increase the number of potential sampling sites for each distance class.

I encountered a few language errors, especially in the introduction and methods, that merit a dedicated verification.

SM: Thanks! We double-checked all the text before resubmitting the manuscript.

Finally, as a little extra: please consider submitting your data to the SoilTemp-database (www.soiltempproject.com) and indeed consider using this database for future analyses, as you suggest yourselves on line 318. This could greatly enhance global coverage of your – and our – analyses. Given the existence of the SoilTemp database, however, I would remove your claim of being the most extensive dataset on high-elevation soil temperatures (L310). This might be true for the particular case of glacier forelands, but most certainly not for high-elevation areas, regardless of your definition of ‘high-elevation’. This made me realize: perhaps the ‘high mountains AND glacier forelands’ in the title is a bit misleading? I hope these comments are helpful,
Kind regards,
Jonas Lembrechts

SM: Many thanks for all your comments, they have been really appreciated. Part of the data have been already uploaded to the SoilTemp dataset a few months ago (all the data originating from the IceCommunities project; for the remaining data the decision is up to the data owners). About the claim on line 310, we modified the “high-elevation” to “proglacial”. Regarding the title, we believe it fully describes the major topic of the manuscript; if the Editor has any alternative suggestion, we would be happy to take it into account.

REVIEWER COMMENTS

Reviewer #1 (Remarks to the Author):

I remain impressed with the quality of the work in this study, and appreciate the lengths to which the authors have gone to in order to address my concerns as it makes one's job as a reviewer feel worthwhile and rewarding. Though some of my suggestions were not followed, the authors clearly articulate the reasons for this, and I am wholly convinced that they have addressed my concerns to the extent to which it is possible or sensible to do so. I have no remaining concerns and am happy for the manuscript to be published in its present form.

Reviewer #2 (Remarks to the Author):

I congratulate the authors on making a wide variety of revisions addressing nearly all of my previous comments along with the comments of the other two referees.

I think some very minor issues remain but these should be very minor revisions.

I think the external validation using the Lambrechts soil dataset is a great addition to the paper, but to be included in the validation it is stated that a threshold is set at 15 images per month (for calculated sfd) "to reduce noise". I understand why this could be beneficial statistically, but could this bias the validation to dry areas or areas with infrequent cloud (this assumes cloud is the main cause of the missing images – it may not be)? If so would this influence the validation and its representativeness? I don't know if this is an issue, but some comment in the text would be useful.

The binary variables of tree cover and permafrost are still not explained well in the text – I think you need to add the responses given in the rebuttal to the text (i.e. originally tree cover was a % variable which was then converted to binary etc)

Finally Figure 2 still has strange labelling of the y axis which varies between sub plots (a to d). It is much better to use whole numbers (as in nearly all the other graphs) and try to be consistent.

One of the features of Extended Figure 2 is that it shows that the underestimation/prediction of soil temperatures at low temperatures which occurs in the CHELSA and TerraClimate is no longer a feature of your model – this is good and worth noting in the text. I am also wondering whether it is more logical to present these graphs the other way round with observed on the x axis and predicted on the y – then the residual represents the bias (overprediction above the line: underprediction below). Currently it is a little counter-intuitive.

I think with these very minor changes the paper should be published.

Reviewer #3 (Remarks to the Author):

I had the pleasure to review an earlier version of this manuscript, and I was happy to see that the authors went to great lengths to incorporate my comments and those from the other two reviewers. I

thus have very few things left to say.

One issue that is left is actually an error on my side: we discovered an issue with the calculation of the monthly temperature data in the published Global maps-paper. We did update already the global maps themselves and are working on a correction to the paper, but I only now realize that I forgot to update the published dataset with the monthly summaries. In short, monthly summaries are shifted approximately two weeks (they go from the 15th of month(i-1) till the 14th of month i. This is mostly fine at the annual mean level as is the main topic of the Global maps-paper, but can have substantial effects at the monthly level. I suggest to get in touch with me to receive the new dataset and rerun your validation analysis, which I very much like, by the way! Here is hoping that it does not change your results too much. My apologies for the extra work that this causes, but it will be safer to use the correct data as you are comparing monthly values with those from macroclimate data sources.

Two more questions about your model:

Why should longwave radiation not be included? I can imagine that it would significantly affect temperature averages, as it is the key radiation component for at least a significant chunk of the day. You are talking in the response to the review of Ilya Maclean about an interaction between tree cover and net radiation, but isn't this also true for vegetation cover as a whole, as we are working with soil temperature here (see also L618-620)? It's not only trees that can block solar radiation, small-stature vegetation can do this as well. If one would model near-surface air temperature, this could potentially be ignored, but not for soil temperature, right? Of course, one can keep adding terms to such a mechanistic model, but it feels a bit incomplete to include a tree cover component and not a short-stature vegetation component when there are only 1.7% of plots with trees. If it cannot or should not be included in the model, at least discuss its importance and lack thereof and reasoning for excluding it, so a reader knows what's up.

L81: I got a bit confused about the proposed correction to add 'spatial': are the fluctuations talked about not temporal? Isn't buffering usually seen as a temporal component (which of course has different magnitude across space)? I do think that the concept of vegetation (and topography) buffering understory and especially soil temperatures is by now well established. Of course, this is not true in all systems, but it is in forests for example.

L92: 'during the snow-free season' is what put me on the wrong foot regarding your sfd-parameter. Perhaps rephrase this in such a way that it is clear that the shoulder seasons – not entirely snow-free – are also included?

L102: reference to the external dataset?

L125: perhaps say: 'downscaled using elevational lapse rate'?

L133: perhaps mention somewhere in the discussion that some parameters, e.g. permafrost, might become more important if datasets with more detail could be included?

L305: temperature is 'warming' faster

L329: perhaps here mention again that microclimate now turns out to be sometimes even warming faster than macroclimate, which of course works against the existing microrefugia. In these particular regions (tropics and/or close to glaciers), climate change risks are amplified due to the faster microclimate warming. I think this is a very important finding!

L577: I have a hard time extracting from the text what the 24 estimates of hourly values are, could you make this clearer, potentially with a scheme of some sorts? Now, this remains a bit of a black box to me.

L604: do you have any reference why we could call a wRMSE of 13.96% 'excellent'? This sounds like a substantial error to me. Or – simply rephrase go 'good'?

L707-710: it's actually rather surprising to find such high correlations between mean and extreme temperatures, given that vegetation cover usually affects extremes yet not the mean, which results in often entirely disconnected mean/max, mean/min and min/max values. Perhaps it's worth a line or two to why this would not be the case in your study system? Perhaps the vegetation cover is relatively homogeneous?

Fig. 3: perhaps also mention in the legend for which latitudinal range you do your predictions?

Kind regards,

Jonas Lembrechts

DETAILED RESPONSE TO THE REVIEWERS' COMMENTS

RESPONSE TO COMMENTS OF REVIEWER #1

I remain impressed with the quality of the work in this study, and appreciate the lengths to which the authors have gone to in order to address my concerns as it makes one's job as a reviewer feel worthwhile and rewarding. Though some of my suggestions were not followed, the authors clearly articulate the reasons for this, and I am wholly convinced that they have addressed my concerns to the extent to which it is possible or sensible to do so. I have no remaining concerns and am happy for the manuscript to be published in its present form.

SM: So many thanks for your kind words, and the appreciation of the efforts made to improve the analyses following your suggestions.

RESPONSE TO COMMENTS OF REVIEWER #2

I congratulate the authors on making a wide variety of revisions addressing nearly all of my previous comments along with the comments of the other two referees.

I think some very minor issues remain but these should be very minor revisions.

I think the external validation using the Lambrechts soil dataset is a great addition to the paper, but to be included in the validation it is stated that a threshold is set at 15 images per month (for calculated sfd) "to reduce noise". I understand why this could be beneficial statistically, but could this bias the validation to dry areas or areas with infrequent cloud (this assumes cloud is the main cause of the missing images – it may not be)? If so would this influence the validation and its representativeness? I don't know if this is an issue, but some comment in the text would be useful.

SM: We thank you for the appreciation of the work done to reply to all your comments and constructive criticism!

We agree that, in principle, this might bias the validation toward the less cloudy areas. Unfortunately, this is a general problem of remote sensing data, as lack of a sufficient number of images can be problematic for many typologies of analysis. Following your suggestion, we now acknowledge in the text the presence of a possible bias in the external validation toward arid / low-cloudiness areas, and we highlight that this choice was needed to obtain a sufficient number of images. Lines 616-618: "In principle, this approach might bias the model toward less cloudy areas, as they generally have a larger number of images per month. Nevertheless, we had to limit analyses to months with a minimum number of images to avoid inaccurate sfd estimates."

The binary variables of tree cover and permafrost are still not explained well in the text – I think you need to add the responses given in the rebuttal to the text (i.e. originally tree cover was a % variable which was then converted to binary etc)

SM: We expanded the text according to your suggestions. Lines 633-634: "Original cover data (i.e., percentage of cell covered by trees) were converted to tree occurrence for all the cells with > 0 tree

cover.” and 636-637 “To convert permafrost extent (expressed as the proportion of a cell that is underlain by permafrost) to occurrence of continuous...”.

Finally Figure 2 still has strange labelling of the y axis which varies between sub plots (a to d). It is much better to use whole numbers (as in nearly all the other graphs) and try to be consistent.

SM: This kind of plots show partial residuals, and as a general “rule”, the scale at which they must be represented is the scale at which variation effectively occurs for a given variable. This helps visualizing the relative effect of single predictors and further evaluating (by-eye) the statistical significance (or lack of significance) of the differences.

We tried nonetheless to modify the Figure following all your suggestions. We:

- *used whole numbers;*
- *put panels a-d on the same scale.*

This slightly reduces the “resolution” in panels c and d (i.e., everything is compressed in a slightly smaller portion of the plotting space), but inter-group differences remain clearly visible.

One of the features of Extended Figure 2 is that it shows that the underestimation/prediction of soil temperatures at low temperatures which occurs in the CHELSA and TerraClimate is no longer a feature of your model – this is good and worth noting in the text. I am also wondering whether it is more logical to present these graphs the other way round with observed on the x axis and predicted on the y – then the residual represents the bias (overprediction above the line: underprediction below). Currently it is a little counter-intuitive.

SM: You are right about the importance of mentioning in the text the “lack of underestimation” (with the new dataset it became a slight overestimation) at low temperatures for our dataset (with respect to TerraClimate and Chelsa). We added this note at lines 176-178: “Our model limited the underestimation of soil temperature at low temperatures that occurs with macroclimatic products (Extended Data Fig. 2d vs 2e-f) and outperformed both...”.

About rotating the plots, we considered your suggestion, i.e. we tried putting the expected temperature on the y-axis.

However, we believe inverting the x-y axes (see below for the plots) would produce less clear results, especially for the Internal validation dataset (only relative to the fit lines in red) for two reasons

1. Conceptual reason: the aim of this analysis was testing the ability of different products (our predictions vs TerraClimate vs Chelsa) in predicting observations, not the opposite (that is, “Observed temperature” is assumed to be “dependent” on the other products). Putting the dependent variable on the y axis is the standard approach of biplots and improves readability.

2. Statistical reason: these plots do not strongly improve the visualization of the patterns because of the way regression lines are estimated. If we keep “Observed temperature” as the dependent variable, slopes are always estimated as $\frac{Cov(Expected, Observed)}{Var(Observed)}$ with $Var(Observed)$ being constant within both a-c and d-f.

Conversely, if we rotate the axes, the “reference” variance changes from plot to plot (e.g., for Chelsea, the slope will be $\frac{Cov(Observed, Chelsea)}{Var(Chelsea)}$, while for TerraClimate $\frac{Cov(Observed, TerraClimate)}{Var(TerraClimate)}$).

Therefore, even if over- or underestimations are easier to see in this way, fit lines are mathematically less comparable.

Finally we highlight that, in all the Fig 2, the dependent variable (Observed temperature) is the y-axis, thus it would be counterintuitive changing the y-axis for panels 2e-2f only

I had the pleasure to review an earlier version of this manuscript, and I was happy to see that the authors went to great lengths to incorporate my comments and those from the other two reviewers. I thus have very few things left to say.

One issue that is left is actually an error on my side: we discovered an issue with the calculation of the monthly temperature data in the published Global maps-paper. We did update already the global maps themselves and are working on a correction to the paper, but I only now realize that I forgot to update the published dataset with the monthly summaries. In short, monthly summaries are shifted approximately two weeks (they go from the 15th of month(i-1) till the 14th of month i. This is mostly fine at the annual mean level as is the main topic of the Global maps-paper, but can have substantial effects at the monthly level. I suggest to get in touch with me to receive the new dataset and rerun your validation analysis, which I very much like, by the way! Here is hoping that it does not change your results too much. My apologies for the extra work that this causes, but it will be safer to use the correct data as you are comparing monthly values with those from macroclimate data sources.

SM: Dear Dr. Lembrechts, we want to thank you for your appreciation of the efforts made to improve the manuscript. In this new version of the manuscript, the external validation was re-run with the corrected SoilTemp database, and Figure 2f and Extended Data Figure 2 were re-drawn. As expected, wR^2 , $wMAE$ and $wRMSE$ all measured an increase in the goodness-of-fit, for both our predictions and the climate products.

Two more questions about your model:

Why should longwave radiation not be included? I can imagine that it would significantly affect temperature averages, as it is the key radiation component for at least a significant chunk of the day. You are talking in the response to the review of Ilya Maclean about an interaction between tree cover and net radiation, but isn't this also true for vegetation cover as a whole, as we are working with soil temperature here (see also L618-620)? It's not only trees that can block solar radiation, small-stature vegetation can do this as well. If one would model near-surface air temperature, this could potentially be ignored, but not for soil temperature, right? Of course, one can keep adding terms to such a mechanistic model, but it feels a bit incomplete to include a tree cover component and not a short-stature vegetation component when there are only 1.7% of plots with trees. If it cannot or should not be included in the model, at least discuss its importance and lack thereof and reasoning for excluding it, so a reader knows what's up.

SM: We fully agree that, in principle, considering herbaceous vegetation can be useful. Unfortunately, to our knowledge there are no global, high-resolution layers of vegetation cover and height. Longwave radiation certainly contributes to soil temperature, but a proper evaluation of its effect and its fine-scale spatial variability relies on information on vegetation. This is especially the case in proglacial plains, where succession and microtopography produce huge changes in vegetation cover and height in relatively small distances (the great part of them are from "not vegetated" to "very scarcely vegetated" to "scarcely vegetated"). Following your suggestion, we expanded the text to discuss this issue (lines 639-641: "Lack of high-resolution, global data on vegetation cover and height hampered the introduction of modelling terms accounting for both longwave radiation and soil shading by herbs and mosses.").

L81: I got a bit confused about the proposed correction to add 'spatial': are the fluctuations talked about not temporal? Isn't buffering usually seen as a temporal component (which of course has different

magnitude across space)? I do think that the concept of vegetation (and topography) buffering understory and especially soil temperatures is by now well established. Of course, this is not true in all systems, but it is in forests for example.

SM: In the literature, the term “microclimatic buffering” has been used to describe both the buffering of fluctuations that occur over time (e.g. De Frenne et al 2021 Global Change Biology), and to describe the extent at which microclimatic heterogeneity arising from landscape topography can buffer impacts of climate change (e.g. Suggit et al 2018 Nature Climate Change). In principle, both effects can be relevant to our study. The term “spatial” was added following the request by I. Maclean, who was probably thinking about the effect described in Suggit et al. 2018. Nevertheless, we fully agree that temporal effects are extremely relevant. To clarify the sentence, we thus 1) removed the term spatial, 2) in our definition we combine the terms used by these two seminal studies and 3) we now cite both references as source.

L92: ‘during the snow-free season’ is what put me on the wrong foot regarding your sfd-parameter. Perhaps rephrase this in such a way that it is clear that the shoulder seasons – not entirely snow-free – are also included?

SM: We rephrased to make it clearer. Lines 92-93: “...during the snow-free season (i.e., when snow cover is strongly reduced or absent) in high mountains and proglacial environments.”.

L102: reference to the external dataset?

SM: Added.

L125: perhaps say: ‘downscaled using elevational lapse rate’?

SM: Modified.

L133: perhaps mention somewhere in the discussion that some parameters, e.g. permafrost, might become more important if datasets with more detail could be included?

SM: We acknowledged the effect that an increased sample size might have on the model result at lines 352-354 (“...might improve the global representation of the soil temperature dynamics, and possibly allow to better define the effect of some parameters (i.e., vegetation, permafrost) on soil temperature in glacier-related environments.”).

L305: temperature is ‘warming’ faster

SM: Modified.

L329: perhaps here mention again that microclimate now turns out to be sometimes even warming faster than macroclimate, which of course works against the existing microrefugia. In these particular regions (tropics and/or close to glaciers), climate change risks are amplified due to the faster microclimate warming. I think this is a very important finding!

SM: Good point! Expanded following your suggestion. Lines 330-334: "...by the end of the century², Worryingly, microclimate is warming even faster than macroclimate in the areas closer to glaciers, with a particularly strong pattern in the tropical and Southern hemisphere mountains (Extended Data Figure 4). In these regions, the probability of long-term persistence in microrefugia is particularly low, as climate change risks are amplified by the faster microclimate warming."

L577: I have a hard time extracting from the text what the 24 estimates of hourly values are, could you make this clearer, potentially with a scheme of some sorts? Now, this remains a bit of a black box to me.

SM: To clarify, we expanded the sentence adding several details, and rephrased it to "The above-detailed procedure was used to obtain 24 estimates of hourly shortwave radiation (i.e., we calculated one estimate for each hour of the 15th day of each month). The hourly estimates were finally summed up to obtain the monthly-averaged daily cumulative shortwave radiation ($\text{MJ m}^{-2} \text{d}^{-1}$)." (lines 584-587).

L604: do you have any reference why we could call a wRMSE of 13.96% 'excellent'? This sounds like a substantial error to me. Or – simply rephrase go 'good'?

SM: Modified to "good". In the previous sentence, we used the term "excellent" because $wR^2 = 0.91$ and $wMAE = 5\%$, but we refrain from excessively optimistic terms.

L707-710: it's actually rather surprising to find such high correlations between mean and extreme temperatures, given that vegetation cover usually affects extremes yet not the mean, which results in often entirely disconnected mean/max, mean/min and min/max values. Perhaps it's worth a line or two to why this would not be the case in your study system? Perhaps the vegetation cover is relatively homogeneous?

SM: We highlight that, in our study system, vegetation cover is frequently scarce and / or homogeneous. It should be considered that we are almost always dealing with high-elevation tundra, mainly composed by small plants. Under these conditions, the disconnection between average and extremes might be less evident, in our opinion. We clarified this issue at lines 724-729: "Monthly mean temperature was highly correlated to temperature extremes (Pearson's $r > 0.91$ for both minimum and maximum temperature). This strong relationship between temperature average and extremes is possibly due to the rather scarce and homogeneous soil cover, which is mostly occupied by sparse vegetation and high-elevation tundra and has a different behaviour from what is observed in forests³⁴. This suggests that mean temperature provides a good representation of the overall pattern within each month."

Fig. 3: perhaps also mention in the legend for which latitudinal range you do your predictions?

SM: Mentioned.

REVIEWERS' COMMENTS

Reviewer #2 (Remarks to the Author):

I have looked at the second round of modifications and I can see that many sensible adjustments have been made in response to my comments, and explanations have been added to the text where required. In the few cases where they have chosen not to follow suggestions, good reasoning is given as to why not. Therefore, I think the paper is now acceptable for publication and congratulate the authors on producing a highly interesting piece of research.

Reviewer #3 (Remarks to the Author):

No further comments, all previous comments have been implemented sufficiently.

Congratulations to the authors!

Kind regards,

Jonas Lembrechts